# Machine learning structure preserving brackets for forecasting irreversible processes

**Kookjin Lee**
School of Computing
and Augmented Intelligence
Arizona State University
Tempe, AZ 85281

**Nathaniel Trask**
Center for Computing Research
Sandia National Laboratories
Albuquerque, NM 87123
`natrask@sandia.gov`

**Panos Stinis**
Pacific Northwest National Laboratory
Richland, WA 99354

## Abstract

Forecasting of time-series data requires imposition of inductive biases to obtain predictive extrapolation, and recent works have imposed Hamiltonian/Lagrangian form to preserve structure for systems with *reversible* dynamics. In this work we present a novel parameterization of dissipative brackets from metriplectic dynamical systems appropriate for learning *irreversible* dynamics with unknown a priori model form. The process learns generalized Casimirs for energy and entropy guaranteed to be conserved and nondecreasing, respectively. Furthermore, for the case of added thermal noise, we guarantee exact preservation of a fluctuation-dissipation theorem, ensuring thermodynamic consistency. We provide benchmarks for dissipative systems demonstrating learned dynamics are more robust and generalize better than either "black-box" or penalty-based approaches.

## 1 Background and previous work

Modeling time-series data as a solution to a dynamical system with learnable dynamics has been shown to be effective in both data-driven modeling for physical systems and traditional machine learning (ML) tasks. Broadly, it has been observed that imposition of physics-based structure leads to more robust architectures which generalize well [1]. On one end of the spectrum of inductive biases, universal differential equations (UDE) [2] assume an a priori known model form, thus imposing the strongest bias. On the other, neural ordinary differential equations (NODEs) [3] assume a completely black-box model form with minimal bias.

Many recent approaches have turned to structure preserving models of reversible dynamics to obtain an inductive bias that lies in between [4, 5, 6, 7, 8]. One may use black-box deep neural networks (DNNs) to learn an energy of a system with unknown model form, while the algebraic structure of Hamiltonian/Lagrangian dynamics provides a flow map which conserves energy. Typically, the learned flow map has symplectic structure so that phase space trajectories are conserved. In classification problems, this mitigates the vanishing/exploding gradient problem and improves accuracy [9]; in physics, this guarantees that extrapolated states are physically realizable [10].

Such approaches are only appropriate for reversible systems lacking friction or dissipation. In the physics literature, the theory of metriplectic dynamical systems provides a generalization of the Poisson brackets of Hamiltonian/Lagrangian mechanics which model not just a conserved energy, but generalized Casimirs such as entropy [11, 12]. Physical systems which can be cast in this framework

35th Conference on Neural Information Processing Systems (NeurIPS 2021).

obtain a number of mimetic properties related to thermodynamic consistency: satisfaction of the first and second laws of thermodynamics and, for closed stochastic systems, a fluctuation dissipation theorem (FDT) that guarantees thermal forcing is balanced exactly by dissipative forces in equilibrium [13]. This FDT property is particularly critical to analyzing rare events in molecular phenomena driven by thermal noise [14].

Classically, metriplectic systems are obtained by deriving a model from first principles and then observing that the system admits a requisite algebraic structure. While effective for a wide range of physical systems [15], the requisite first principles modeling may be restrictively complex for a general system, particularly for multiscale problems involving time history. *In this work we reverse the process by assuming our time-series data has been generated by a metriplectic system and then inferring the requisite algebraic objects using a training strategy similar to that used in NODEs.* This presents several technical challenges. First, dissipative systems typically have non-observable states (i.e. internal entropy or temperature) which may not be measured. Second, the metriplectic algebraic structure is particularly restrictive, requiring discovery of matrices with carefully designed null-spaces to separate reversible and irreversible components of the dynamics.

**Anticipated impact:** Finally, we note that this work is an important first step toward handling more complicated dissipative chaotic systems ubiquitous to science and engineering problems. For example, for chaotic systems the "butterfly effect" causes arbitrarily small perturbations in initial data to exponentially diverge, and it is only possible to provide long-term forecasting by learning a corresponding "strange attractor" whose latent dimension is governed by the dissipative structure [16, 17]. In reduced order-modeling, many have looked toward data-driven means of fitting dynamics to latent representations of solution space, with Hamiltonian structure particularly useful for finding long-time accurate surrogates[18, 19]. In this situation as well, structure-preserving treatment of dissipation is critical to account for entropic/memory effects which emerge from coarse-graining [20]. Another success of reversible structure-preserving ML is in robotic control [21]. Again these models fail to account for friction due to wear, which is inevitable in realistic applications. There are also problems where we have access to time-series and no guaranteed way of modeling from first principles that can account for all the important mechanisms e.g. system-identification in biology [22, 23], the study of cascading failures for realistic power grid models [24], social dynamics [25], and accounting for memory effects in empirical eigenfunction expansions of turbulent flows [26]. All these cases can benefit by a structure-preserving method for identifying dynamics.

## 2 Related work

**Neural ordinary differential equations**   As noted, learning time-continuous dynamics in the form of a system of ODEs is an active topic with seminal works including [27, 9, 28, 3, 29]. There have been many follow-up studies to enhance neural ODEs in different aspects, e.g., enhancing the expressivity of neural ODEs by augmenting extra dimensions in state variables [30], checkpoint methods to mitigate numerical instability and to enhance memory efficiency [31, 32, 33], allowing network parameters to evolve over time together with hidden states [34, 35], and spectrally approximating dynamics by using a set of orthogonal polynomials [36]. Applications of NODEs for learning complex physical processes (e.g., turbulent flow) can be found in [37, 38, 39].

**Structure preserving neural networks**   A thorough accounting of works embedding structure-preservation into neural networks include pioneering works for Hamiltonian neural networks [4, 40], followed by development of Lagragian neural networks [41, 5] and neural networks that mimic the action of symplectic integrators [6, 7, 8]. More recently, there has been efforts to add physical invariance to learned dynamics models, e.g., time-reversal symmetry [42]. Works pursuing related but distinct spatial-compatibility related to conservation structure other than geometric integration include: graph architectures with associated a data-driven graph exterior calculus [43], solving optimization problems with conservation constraint in latent space [44], and adding conservation constraints as a penalty in training loss [45]. The closest work to our approach is in [46], which proposed a time integrator that leverages the GENERIC (general equation for the nonequilibrium reversible–irreversible coupling) formalism to impose the structure, but enforces the degeneracy condition as penalty terms in the training loss objective. We will provide results demonstrating that a penalty approach is insufficient to guarantee preservation of metriplectic structure.

## 3 Theory and fundamentals

We consider the GENERIC formalism as a particular metriplectic framework amenable to parameterization. Consider time series data $\mathcal{D} = \{(t_i, \boldsymbol{x}(t_i))\}_{i=1}^N$, where the state $\boldsymbol{x}_i = \boldsymbol{x}(t_i) \subset \mathbb{R}^d$ has a known initial condition $\boldsymbol{x}_0$. In GENERIC, it is assumed that an observable $A(\boldsymbol{x})$ evolves under the gradient flow

$$\frac{\mathrm{d}A}{\mathrm{d}t} = \{A, E\} + [A, S] \tag{1}$$

where $E$ and $S$ denote generalized energy and entropy, $\{\cdot, \cdot\}$ denotes a Poisson bracket, and $[\cdot, \cdot]$ denotes an irreversible bracket. The Poisson bracket is given in terms of a skew-symmetric Poisson matrix $L$ and the irreversible bracket is given in terms of a symmetric positive semi-definite friction matrix $M$,

$$\{A, B\} = \frac{\partial A}{\partial \boldsymbol{x}} L \frac{\partial B}{\partial \boldsymbol{x}}, \quad \text{and} \quad [A, B] = \frac{\partial A}{\partial \boldsymbol{x}} M \frac{\partial B}{\partial \boldsymbol{x}}.$$

A system governed by Eq. 1 is a GENERIC system if the following degeneracy conditions hold

$$L \frac{\partial S}{\partial \boldsymbol{x}} = 0, \quad \text{and} \quad M \frac{\partial E}{\partial \boldsymbol{x}} = 0. \tag{2}$$

Taking $A = \boldsymbol{x}$ in Eq. (1) provides the evolution of $\boldsymbol{x}$

$$\frac{\mathrm{d}\boldsymbol{x}}{\mathrm{d}t} = L \frac{\partial E}{\partial \boldsymbol{x}} + M \frac{\partial S}{\partial \boldsymbol{x}}. \tag{3}$$

**Remark 3.1 (Hamiltonian dynamics)** *For canonical coordinates $\boldsymbol{x} = [q, p]^\mathsf{T}$, and canonical Poisson matrix $L = \begin{bmatrix} 0 & 1 \\ -1 & 0 \end{bmatrix}$, and $M = \mathbf{0}$, Eq. (3) recovers Hamiltonian dynamics.*

**Remark 3.2 (First and second laws of thermodynamics)** *Taking $A = E$ and $A = S$, we obtain $\frac{\mathrm{d}E}{\mathrm{d}t} = 0$ and $\frac{\mathrm{d}S}{\mathrm{d}t} \geq 0$, respectively. This follows easily by application of the degeneracy conditions and noticing $\{A, A\} = 0$, $[A, A] \geq 0$.*

**Remark 3.3 (Fluctuation dissipation theorem)** *Introducing thermal noise to Eq. (3) provides the stochastic differential equation (SDE)*

$$\mathrm{d}\boldsymbol{x}_t = \left( L \frac{\partial E}{\partial \boldsymbol{x}} + M \frac{\partial S}{\partial \boldsymbol{x}} + k_B \frac{\partial}{\partial \boldsymbol{x}} \cdot M \right) \mathrm{d}t + \sqrt{2 k_B M} \mathrm{d}W_t, \tag{4}$$

*where $\sqrt{M}$ denotes the Cholesky factor of $M$, $k_B$ is a Boltzmann constant, and $\mathrm{d}W_t$ is a Wiener process. The equilibrium statistics of this SDE reach a stationary distribution under appropriate conditions [13].*

## 4 Parameterization of bracket structure

We now introduce a parameterization of the dissipative and reversible brackets that exactly satisfies the degeneracy conditions described in Section 3, and review the penalty approach from [46] which imposes degeneracy conditions via soft constraints. Our approach is motivated by the work in [47] which we summarize in Sections 4.1–4.3. For the remainder, we adopt the Einstein summation convention.

First, we parameterize the energy and the entropy as neural networks, i.e., $E(\boldsymbol{x}) \approx E_\phi(\boldsymbol{x})$ and $S(\boldsymbol{x}) \approx S_\varphi(\boldsymbol{x})$, where $\phi$ and $\varphi$ are weights and biases for the neural networks $E$ and $S$ respectively.

### 4.1 Parameterizing skew-symmetric reversible dynamics

The reversible dynamics are characterized by a skew-symmetric Poisson bracket,

$$\{A, B\} = \xi_{\alpha\beta\gamma} \frac{\partial A}{\partial x_\alpha} \frac{\partial B}{\partial x_\beta} \frac{\partial S}{\partial x_\gamma},$$

where $\xi_{\alpha\beta\gamma}$ is an skew-symmetric 3d tensor. To enforce the anti-symmetry exactly, we consider a generic 3 tensor $\tilde{\xi}_{\alpha\beta\gamma}$ with learnable entries and apply the following skew-symmetrization trick.

$$\xi_{\alpha\beta\gamma} = \frac{1}{3!} \left( \tilde{\xi}_{\alpha\beta\gamma} - \tilde{\xi}_{\alpha\gamma\beta} + \tilde{\xi}_{\beta\gamma\alpha} - \tilde{\xi}_{\beta\alpha\gamma} + \tilde{\xi}_{\gamma\alpha\beta} - \tilde{\xi}_{\gamma\beta\alpha} \right).$$

The reversible part may then be written as $\{x, E\} = \xi_{\alpha\beta\gamma} \frac{\partial x}{\partial x_\alpha} \frac{\partial E}{\partial x_\beta} \frac{\partial S}{\partial x_\gamma}$ and the reversible dynamics are given by

$$\left( \frac{\mathrm{d}x_\alpha}{\mathrm{d}t} \right)_{\mathrm{r}} = \xi_{\alpha\beta\gamma} \frac{\partial E}{\partial x_\beta} \frac{\partial S}{\partial x_\gamma}.$$

## 4.2 Parameterizing symmetric irreversible dynamics

Next, we parameterize the irreversible dynamics via the bracket,

$$[A, B] = \zeta_{\alpha\beta,\mu\nu} \frac{\partial A}{\partial x_\alpha} \frac{\partial E}{\partial x_\beta} \frac{\partial B}{\partial x_\mu} \frac{\partial E}{\partial x_\nu},$$

where

$$\zeta_{\alpha\beta,\mu\nu} = \Lambda_{\alpha\beta}^m D_{mn} \Lambda_{\mu\nu}^n.$$

Here, $\Lambda$ and $D$ are skew-symmetric and symmetric positive semi-definite matrices, respectively, such that

$$\Lambda_{\alpha\beta}^m = -\Lambda_{\beta\alpha}^m, \quad \text{and} \quad D_{mn} = D_{nm}.$$

Again, the skew-symmetry and the symmetric positive semi-definiteness can be achieved by the parameterization tricks

$$\Lambda = \frac{1}{2}(\tilde{\Lambda} - \tilde{\Lambda}^{\mathsf{T}}), \quad \text{and} \quad D = \tilde{D}\tilde{D}^{\mathsf{T}},$$

where $\tilde{\Lambda}$ and $\tilde{D}$ are matrices with learnable entries. Finally, the irreversible part may be written as $[x, S] = \zeta_{\alpha\beta,\mu\nu} \frac{\partial x}{\partial x_\alpha} \frac{\partial E}{\partial x_\beta} \frac{\partial S}{\partial x_\mu} \frac{\partial E}{\partial x_\nu}$ and the irreversible part of the dynamics is given by

$$\left( \frac{\mathrm{d}x_\alpha}{\mathrm{d}t} \right)_{\mathrm{irr}} = \zeta_{\alpha\beta,\mu\nu} \frac{\partial E}{\partial x_\beta} \frac{\partial S}{\partial x_\mu} \frac{\partial E}{\partial x_\nu}.$$

## 4.3 Degeneracy conditions

With the above parameterizations the degeneracy conditions described in Eq. (2) may be easily verified by direct calculation following the definition of the brackets and the symmetry/skew-symmetry conditions.

$$\{x, S\} = \frac{\partial x}{\partial x} L \frac{\partial S}{\partial x} = \xi_{\alpha\beta\gamma} \frac{\partial x}{\partial x_\alpha} \frac{\partial S}{\partial x_\beta} \frac{\partial S}{\partial x_\gamma} = \xi_{\alpha\beta\gamma} \frac{\partial S}{\partial x_\beta} \frac{\partial S}{\partial x_\gamma} = 0,$$

and

$$[x, E] = \frac{\partial x}{\partial x} M \frac{\partial E}{\partial x} = \zeta_{\alpha\beta,\mu\nu} \frac{\partial x}{\partial x_\alpha} \frac{\partial E}{\partial x_\beta} \frac{\partial E}{\partial x_\mu} \frac{\partial E}{\partial x_\nu} = \zeta_{\alpha\beta,\mu\nu} \frac{\partial E}{\partial x_\beta} \frac{\partial E}{\partial x_\mu} \frac{\partial E}{\partial x_\nu} = 0.$$

## 4.4 Alternative parameterization – penalty-based method

An alternative strategy to incorporate GENERIC structure is to enforce the degeneracy condition by soft penalty as advocated in [46]. In this approach, $E$, $S$, $L$, and $M$, may be approximated independently of each other. Again, $E$ and $S$ are parameterized as neural networks ($E_\phi$ and $S_\varphi$), and $L$ and $M$ are parameterized as skew-symmetrizations/symmetrizations of matrices $\pi$ and $\rho$ with learnable entries as follows

$$L_{\pi} = \frac{1}{2} \left( \pi - \pi^{\mathsf{T}} \right) \quad \text{and} \quad M_{\rho} = \rho\rho^{\mathsf{T}}.$$

With this parameterization, the degeneracy conditions are simply enforced by minimizing two penalty terms, $\left\| L_{\pi} \frac{\partial E_\phi}{\partial x} \right\|$ and $\left\| M_{\rho} \frac{\partial S_\varphi}{\partial x} \right\|$. We stress that this penalty will be enforced only to within optimization error.

If we write a system of neural ODEs as $\frac{\partial x}{\partial t} = f_\Theta$, where $\Theta$ consists of learnable parameters, then Table 1 summarizes the components comprising $f_\Theta$ for black-box NODE, the penalty-base method, and GENERIC NODE (GNODE).

Table 1: Model summary

| | NODE | Penalty | GNODE |
|---|---|---|---|
| $\boldsymbol{f}_\Theta$ | $\boldsymbol{f}_\Theta = \boldsymbol{f}_\theta$ | $\boldsymbol{f}_\Theta = L_{\boldsymbol{\pi}}\frac{\partial E_\phi}{\partial \boldsymbol{x}} + M_{\boldsymbol{\rho}}\frac{\partial S_{\boldsymbol{\varphi}}}{\partial \boldsymbol{x}}$ | $\boldsymbol{f}_\Theta = \{\boldsymbol{x}, E\} + [\boldsymbol{x}, S]$ |
| Components | black-box MLP | $E_\phi$ and $S_{\boldsymbol{\varphi}}$ (MLPs) 
 $L_{\boldsymbol{\pi}}$ and $M_{\boldsymbol{\rho}}$ (2-tensor) | $E_\phi$ and $S_{\boldsymbol{\varphi}}$ (MLPs) 
 $\xi$ (3-tensor), $\Lambda$ and $D$ (2-tensor) |
| $\Theta$ | $\Theta = \theta$ | $\Theta = \{\phi, \boldsymbol{\varphi}, \boldsymbol{\pi}, \boldsymbol{\rho}\}$ | $\Theta = \{\phi, \boldsymbol{\varphi}, \xi, \Lambda, D\}$ |

## 5 Experiments

In this section, we assess the performance of the three parameterizations of the ODE dynamics which apply progressively more stringent priors. We implement the algorithms in PYTHON 3.6.5, NUMPY 1.16.2, and PYTORCH 1.7.1 [48]. For the time integrator, we use a PYTORCH implementation of differentiable ODE solvers, TorchDiffEq [3]. All experiments are performed on MACBOOK PRO with 2.9 GHz i9 CPU and 32 GB memory.

### 5.1 Dataset and training

The states $\boldsymbol{x}$ of GENERIC systems may generally be partitioned between "observable" states (e.g., position and momentum variables) denoted by $\boldsymbol{x}^{\mathrm{o}}$ and "non-observable" states (e.g., entropy, configuration variables, etc) denoted by $\boldsymbol{x}^{\mathrm{u}}$, i.e., $\boldsymbol{x} = [\boldsymbol{x}^{\mathrm{o}\mathsf{T}}, \boldsymbol{x}^{\mathrm{u}\mathsf{T}}]^{\mathsf{T}}$. We assume that training data is only available for the observable states, with the non-observable states functioning as hidden variables during training. For each benchmark problem, we take as manufactured training data a single trajectory of observable states obtained by integrating a reference ODE with known GENERIC structure from a known initial condition. We then split the sequence into three segments, $[0, t_{\mathrm{train}}]$, $(t_{\mathrm{train}}, t_{\mathrm{val}}]$, and $(t_{\mathrm{val}}, t_{\mathrm{test}}]$ for training, validation, and test such that $0 < t_{\mathrm{train}} < t_{\mathrm{val}} < t_{\mathrm{test}}$.

We employ mini-batching to train all three considered architectures. Each mini-batch consists of multiple short sequences of length $L$ whose initial conditions are randomly chosen from $[0, t_{\mathrm{train}}]$. To train "black-box" neural ODEs, we simply use a stochastic gradient descent (SGD) optimizer to update the network weights and biases using the mini-batches on the observable states, $\{\boldsymbol{x}^{\mathrm{o}}_\ell, \boldsymbol{x}^{\mathrm{o}}_{\ell+1}, \ldots, \boldsymbol{x}^{\mathrm{o}}_{\ell+L-1}\}$.

As opposed to the black-box neural ODEs, training the penalty-based approach and the GENERIC approach requires data to impose mini-batch initial conditions on non-observable states, i.e., $\{\boldsymbol{x}_\ell, \boldsymbol{x}_{\ell+1}, \ldots, \boldsymbol{x}_{\ell+L-1}\}$ with $\boldsymbol{x}_\ell = [\boldsymbol{x}^{\mathrm{o}}_\ell, \boldsymbol{x}^{\mathrm{u}}_\ell]^{\mathsf{T}}$, where $\{\boldsymbol{x}^{\mathrm{u}}_\ell\}$ are unavailable. To address this issue, we propose a training strategy that alternately updates the model parameters and infers the non-observable states. We start with a guess for the non-observable states. We then alternate between (1) updating the model parameters using SGD while fixing the current non-observable states and (2) updating the non-observable states by solving an initial value problem using the most recent model.

---

**Algorithm 1:** Neural ODE training

1   Initialize $\Theta$
2   **for** $(i = 0;\ i < n_{\max};\ i = i + 1)$ **do**
3      Sample initial points $\{\boldsymbol{x}^{\mathrm{o}}_{\ell(k)}\}_{k=1}^{N_{\mathrm{b}}}$, where $\ell(k) \in [0, t_{\mathrm{train}} - L - 1]$ for $k = 1, \ldots, N_{\mathrm{b}}$
4      $\tilde{\boldsymbol{x}}^{\mathrm{o}}_{\ell(k)+1}, \ldots, \tilde{\boldsymbol{x}}^{\mathrm{o}}_{\ell(k)+L} = \mathrm{ODESolve}(\boldsymbol{x}^{\mathrm{o}}_{\ell(k)}, \boldsymbol{f}_\Theta, t_1, \ldots, t_L)$ for $k = 1, \ldots, N_{\mathrm{b}}$
5      Compute loss: $\mathcal{L}(\boldsymbol{x}^{\mathrm{o}}_{\ell(k)+m}, \tilde{\boldsymbol{x}}^{\mathrm{o}}_{\ell(k)+m})$
6      Update $\Theta$ via SGD

---

For ODESolve, we use the Dormand–Prince method (dopri5) [49] with relative tolerance $10^{-5}$ and absolute tolerance $10^{-6}$. The loss function $\mathcal{L}$ measures the discrepancy between the ground truth states and approximate states via mean absolute errors, and the network weights and biases are updated using Adamax [50] with an initial learning rate 0.01.

In the following, we test the proposed algorithms with two benchmark problems: a damped nonlinear oscillator and two gas containers problems. Data for all considered benchmark problems can be found in [51].

**Algorithm 2:** Penalty or GENERIC training

1  Initialize $\Theta$ and $\{\boldsymbol{x}_0^u, \ldots, \boldsymbol{x}_{t_{\text{train}}}^u\}$
2  Construct a dataset as $\boldsymbol{x}_i = [\boldsymbol{x}_i^{o\mathsf{T}}, \boldsymbol{x}_i^{u\mathsf{T}}]^\mathsf{T}$, for $i = 0, \ldots, t_{\text{train}}$
3  **for** $(i = 0;\ i < n_{\max};\ i = i+1)$ **do**
4     Sample initial points $\{\boldsymbol{x}_{\ell(k)}\}_{k=1}^{N_b}$, where $\ell(k) \in [0, t_{\text{train}} - L - 1]$ for $k = 1, \ldots, N_b$
5     $\tilde{\boldsymbol{x}}_{\ell(k)+1}, \ldots, \tilde{\boldsymbol{x}}_{\ell(k)+L} = \text{ODESolve}(\boldsymbol{x}_{\ell(k)}, \boldsymbol{f}_\Theta, t_1, \ldots, t_L)$ for $k = 1, \ldots, N_b$
6     Compute loss: $\mathcal{L}(\boldsymbol{x}_{\ell(k)+m}^o, \tilde{\boldsymbol{x}}_{\ell(k)+m}^o)$
7     Update $\Theta$ via SGD
8     **if** $i \bmod n_{\text{update}} == 0$ **then**
9        $\tilde{\boldsymbol{x}}_1, \ldots, \tilde{\boldsymbol{x}}_{t_{\text{train}}} = \text{ODESolve}(\boldsymbol{x}_0, \boldsymbol{f}_\Theta, t_1, \ldots, t_{\text{train}})$
10       Update a dataset as $\boldsymbol{x}_i = [\boldsymbol{x}_i^{o\mathsf{T}}, \tilde{\boldsymbol{x}}_i^{u\mathsf{T}}]^\mathsf{T}$, for $i = 0, \ldots, t_{\text{train}}$

## 5.2  Damped nonlinear oscillator

As a first benchmark problem, we consider a damped nonlinear oscillator which exhibits a natural GENERIC structure:

$$\frac{\mathrm{d}q}{\mathrm{d}t} = \frac{p}{m}, \qquad \frac{\mathrm{d}p}{\mathrm{d}t} = k\sin(q) - \gamma p, \qquad \frac{\mathrm{d}S}{\mathrm{d}t} = \frac{\gamma q^2}{mT}, \tag{5}$$

where $(q, p)$ denote the position and momentum of the particle, and $S$ is the entropy of the surrounding thermal bath. The constant parameters $m$, $\gamma$, and $T$ represent the mass of the particle, the damping rate, and the constant temperature of the thermal bath. The total energy of the GENERIC system is

$$E(q, p, S) = H(q, p) + TS = \frac{p^2}{2m} - k\cos(q) + TS,$$

where $H(q, p)$ is the Hamiltonian of the particle (the sum of the kinetic and the potential energy).

In this benchmark problem, the observable states consist of the position and the momentum variables, i.e., $\boldsymbol{x}^o = [q, p]^\mathsf{T}$. We consider a single non-observable variable, i.e., $\boldsymbol{x}^u = s$. Now, our goal is to learn a system of ODEs that conforms the GENERIC structure described in Section 4 and infer the non-observable variable via Algorithm 2. That is, for GNODE, we model $E_\phi$ and $S_\varphi$ to take $\boldsymbol{x} = [q, p, s]^\mathsf{T}$ as an input.

For black-box NODEs, we have tested MLPs with combinations of $\{2, 3, 4\}$ hidden layers with $\{5, 10, 15\}$ neurons and observed the best result with an MLP with 4 hidden layers with 5 neurons in each layer and hyperbolic tangent (Tanh) activation function. For the penalty-based approach, we have tested MLPs with combinations of $\{2, 3, 4\}$ hidden layers with $\{5, 10, 15\}$ and Tanh for parameterizing $E_\phi$ and $S_\varphi$ and observed the best results with MLPs with 3 hidden layers with 5 neurons in each layer. The $3 \times 3$ learnable entries for $L_\pi$ and $M_\rho$ are considered. We add the penalty terms (see Section 4.4) that are weighted by $10^{-4}$ to the main loss objective. We have also tested $10^{-2}$ and $10^{-6}$ for weighting, which resulted in significant misfit in data or inconsistency in physics (i.e., failure to enforce the degeneracy conditions). Lastly, for the GENERIC approach, we have tested MLPs with combinations of $\{0, 1, 2\}$ layers with $\{5, 10\}$ neurons for parameterizing $E_\phi$ and $S_\varphi$ and we have observed the best results with an MLP with 1 hidden layer with 10 neurons and Tanh for parameterizing $E_\phi$, and a linear layer for parameterizing $S_\varphi$. Then, we use $3 \times 3 \times 3$ skew-symmetric tensor to parameterize $\xi$, $3 \times 3$ skew-symmetric tensor to parameterize $\Lambda$, and $3 \times 1$ tensor, $d$, to parameterize $D$, i.e., $D = dd^\mathsf{T}$. For initializing layers in MLPs, we use the PYTORCH default uniform distribution and, for initializing learnable entries, we initialize them with unit normal distribution. We initialize the non-observable variable as $\boldsymbol{x}_\ell^u = s_\ell = t_\ell$ (i.e., setting it to be monotonically increasing) in Line 1 of Algorithm 2.

The dataset consists of a sequence of 180,000 timesteps with $t_{\text{final}} = 180$ (in second) and step size $\Delta t = 0.001$. We then split the dataset into training, validation, and testing sets such that $t_{\text{train}} = 20$, $t_{\text{val}} = 40$, and $t_{\text{test}} = 180$. Each mini-batch consists of $N_b = 20$ subsequences of length $L = 120$. The maximum training step is set as $n_{\max} = 30000$ and the update is performed at every $n_{\text{update}} = 500$ training steps (in Algorithm 2).

In the experiment, we consider $m = k = T = 1$, and $\gamma = 0.01$. The initial condition is given as $\boldsymbol{x}^0 = [2, 0, 0]^\mathsf{T}$, where the initial condition for the non-observable variable is arbitrarily set. Results

of the comparison are given in Figure 1. The results are obtained from 5 independent runs (i.e., 5 different random seeds). The plots of $\frac{dS}{dt}$ reveal that the soft enforcement of constraints in the penalty formulation leads to negative entropy production of large magnitude, while the GNODE approach enforces by construction $\frac{dS}{dt} \geq 0$. When extrapolating well beyond the training time interval, both black-box NODE and the penalty approach have a large standard deviation in the predicted $H$. GNODE in contrast learns a nearby entropy which consistently dissipates the correct amount of energy.

## 5.3 Two gas containers

The second benchmark problem considers two (ideal) gas containers, separated by a moving wall, exchanging heat and volume. Here, we are interested in the position and the momentum of the separating wall, i.e., $x^{\mathrm{o}} = [q, p]^{\mathsf{T}}$. This problem possesses a highly nonlinear expression for the entropy [51]:

$$\frac{\mathrm{d}q}{\mathrm{d}t} = \frac{p}{m}, \qquad \frac{\mathrm{d}p}{\mathrm{d}t} = \frac{2}{3}\left(\frac{E_1}{p} - \frac{E_2}{2L_g - p}\right),$$

$$\frac{\mathrm{d}S_1}{\mathrm{d}t} = \frac{9N^2 k_B^2 \alpha}{4E_1}\left(\frac{1}{E_1} - \frac{1}{E_2}\right), \qquad \frac{\mathrm{d}S_2}{\mathrm{d}t} = -\frac{9N^2 k_B^2 \alpha}{4E_1}\left(\frac{1}{E_1} - \frac{1}{E_2}\right),$$

where $(q, p)$ denote the position and momentum of the separating wall and $S_1$ and $S_2$ are the entropies of the two subsystems. The constants $m$ denotes the mass of the wall, $2L_g$ is the total length of the two containers. Following [51], we set $Nk_B = 1$, which fixes a characteristic macroscopic unit of

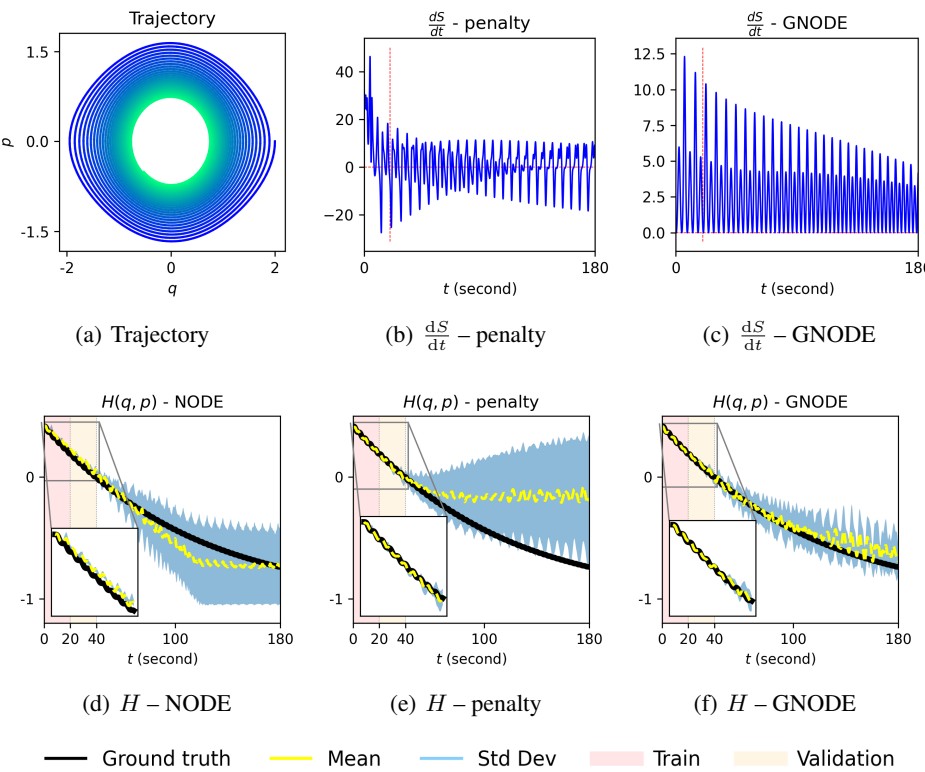

Figure 1: For the damped nonlinear oscillator, the physical entropy may be evaluated via the formula $E = H + TS$. While all three methods fit training data reasonably well, NODE and the penalty approach rapidly deviate. An inspection of $\frac{dS}{dt}$ for the penalty method shows that the soft penalty is insufficient to ensure compatibility with the second law. GNODE is able to consistently learn an entropy $S$ which closely tracks the physical entropy $(E - H)/T$.

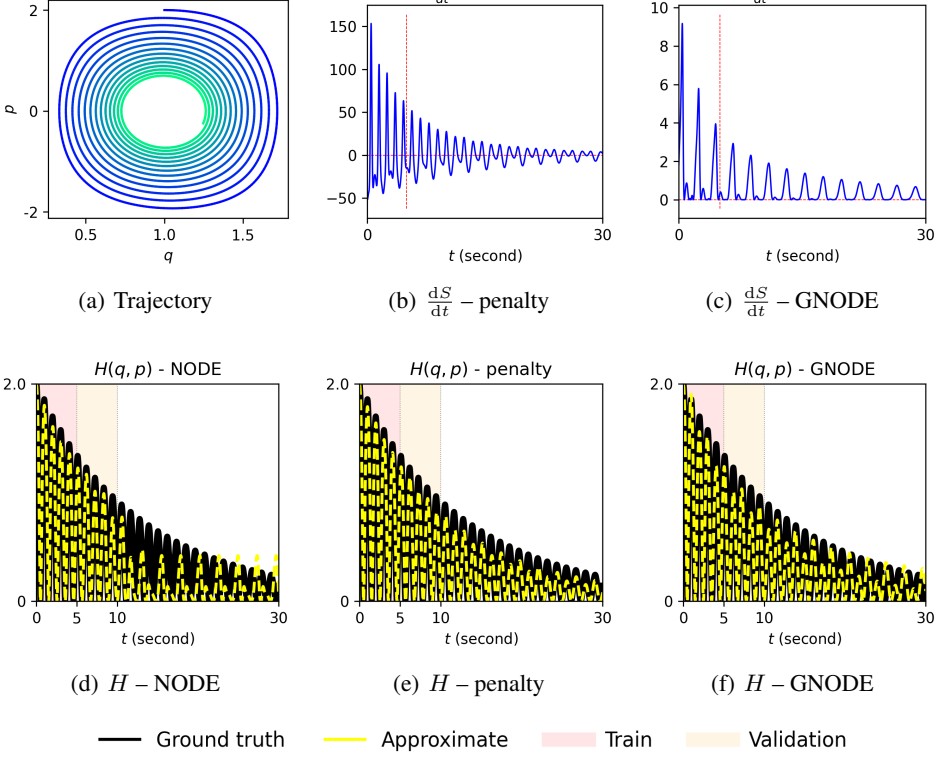

(a) Trajectory      (b) $\frac{\mathrm{d}S}{\mathrm{d}t}$ − penalty      (c) $\frac{\mathrm{d}S}{\mathrm{d}t}$ − GNODE

(d) $H$ − NODE      (e) $H$ − penalty      (f) $H$ − GNODE

—— Ground truth    —— Approximate    ▨ Train    ▨ Validation

Figure 2: For the two gas cylinder problem the network must learn a significantly more complex entropy-energy relationship than in the damped oscillator. The results however are similar; GNODE provides remarkable improvement in forecasting due to its faithful reproduction of the second law of thermodynamics *(top row, center + right)*.

entropy, and $\alpha = 0.5$. The internal energies of the two subsystems has the relationship with the associated entropies and volumes via the Sackur–Tetrode equation for ideal gases such that

$$\frac{S_i}{N k_B} = \ln \left[ \hat{c} V_i (E_i)^{3/2} \right], \quad i = 1, 2,$$

where $\hat{c}$ is a constant to ensure the argument of the logarithm dimensionless (set as $\hat{c} = 102.25$). The total energy is given by

$$E(q, p, S_1, S_2) = H(q, p) + E_1 + E_2 = \frac{p^2}{2m} + E_1 + E_2.$$

Again, the observable states consist of the position and the momentum variables, i.e., $\boldsymbol{x}^{\mathrm{o}} = [q, p]^{\mathsf{T}}$. We consider two non-observable variables, i.e., $\boldsymbol{x}^{\mathrm{u}} = [s_1, s_2]^{\mathsf{T}}$. This problem is more challenging as the dynamics of the observable variables strongly depends on the dynamics of the non-obesrvable variables. Now we train GNODE to learn a system of ODEs that conforms the GENERIC structure described in Section 4 and infer the non-observed variables via Algorithm 2. Again, for GNODE, we model $E_\phi$ and $S_\varphi$ to take $\boldsymbol{x} = [q, p, s_1, s_2]^{\mathsf{T}}$ as an input.

For black-box NODEs, we consider an MLP with 4 hidden layers with 5 neurons in each layer and Tanh as nonlinearity. For the penalty-based approach, we consider MLPs with 3 hidden layers with 5 neurons in each layer and Tanh for parameterizing $E_\phi$ and $S_\varphi$, and $3 \times 3$ learnable entries for $L_{\boldsymbol{\pi}}$ and $M_{\boldsymbol{\rho}}$. We add the penalty terms that are weighted by $1e - 4$ to the main loss objective. Lastly, for the GENERIC approach, we consider an MLP with 2 hidden layer with 5 neurons and Tanh for parameterizing $E_\phi$, and 1 hidden layer with 5 neurons and Tanh for parameterizing $S_\varphi$. Then, we use $3 \times 3 \times 3$ skew-symmetric tensor to parameterize $\xi$, $4 \times 4$ skew-symmetric tensor to parameterize $\Lambda$,

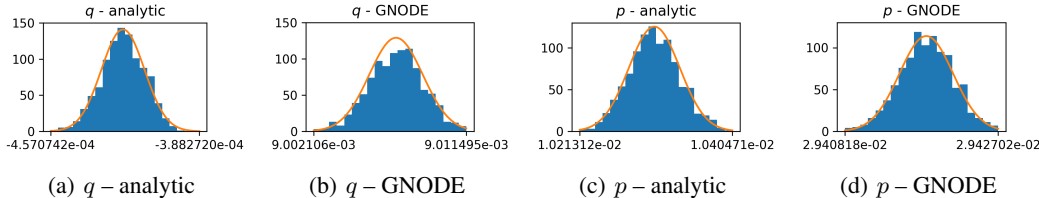

| (a) $q$ – analytic | (b) $q$ – GNODE | (c) $p$ – analytic | (d) $p$ – GNODE |

Figure 3: Distribution of $q$ and $p$ at $t_{\text{final}} = 80$ for the SDE (4). The standard deviations of $(q, p)$ for baseline and GNODE are $(9.9684 \times 10^{-6}, 3.1971 \times 10^{-5})$ and $(1.6436 \times 10^{-6}, 3.2972 \times 10^{-6})$, respectively.

and $4 \times 1$ tensor, $d$, to parameterize $D$, i.e., $D = dd^{\mathsf{T}}$. For initializing layers in MLPs, we use the PYTORCH default uniform distribution and, for initializing learnable entries, we initialize them with unit normal distribution. The non-observable variables are initialized as $s_{1,\ell} = s_{2,\ell} = t_\ell$.

The dataset consists of a sequence of 30,000 timesteps with $t_{\text{final}} = 30$ seconds and step size $\Delta t = 0.001$. We then split the dataset into training, validation, and testing subsets such that $t_{\text{train}} = 5$, $t_{\text{val}} = 10$, and $t_{\text{test}} = 30$. Each mini-batch consists of $N_{\text{b}} = 20$ subsequences of length $L = 40$. The maximum training step is set as $n_{\text{max}} = 50000$ and the update is performed at every $n_{\text{update}} = 500$ training steps (in Algorithm 2).

We set $m = L_g = 1$ and the initial condition is given as $\boldsymbol{x}^0 = [1, 2, 0, 0]$. Again, the initial condition for the non-observable variables are set arbitrarily. The results are depicted in Figure 2 and demonstrate similar results to the damped oscillator. We depict the results from the best performing instances for each model out of five independent runs. Both NODE and the penalty method provide inaccurate forecasting beyond the training set, and the penalty method can be seen to generate negative entropy violating the second law of thermodynamics, while $\frac{dS}{dt} \geq 0$ holds strictly for GNODE.

### 5.4 Stochastic damped harmonic oscillator

We finally fit a GNODE model to the system considered Section 5.2 and use the learned $E$, $S$, $M$ and $L$ in the right hand side of the SDE in (4), and compare as a baseline to using instead the analytical $E$, $S$, $M$, and $L$ from (5). This amounts to driving both the true system and data-driven dynamics with thermal noise which exactly balances the dissipation, and requires the FDT to hold to realize stationary statistics.

In this experiment, we consider the damped nonlinear oscillator with $m = 10$, and $\gamma = 0.16$ and the same neural network architecture considered in Section 5.2. We use a sequence of 80,000 timesteps ($t_{\text{final}} = 80$ and $\Delta t = 0.0001$) to train the neural network. We use the same training strategy that is used in Section 5.2 (Algorithm 2); the only difference is that we use the mini-batch of size $N_{\text{b}} = 40$.

For Figure 3 we obtain statistics from solving (4) with $t_{\text{final}} = 80$ of Euler–Maruyama [52] with step size $\Delta t = 0.001$. The mean of the resulting SDE solutions show substantial deviation, consistent with the fact that training is performed only on deterministic training data and not the SDE, which is compounded by nonlinearities in the data. However, the standard deviation of the distribution shows good agreement, suggesting that the FDT enforces thermodynamically consistent energy budget between dissipation and stochastic forcing.

## 6 Conclusions

We have constructed a generalization of structure preserving networks for reversible dynamics to handle dissipative processes. Unlike the reversible case, the bracket structure requires a much more careful treatment of degeneracy conditions to ensure compatibility with the first and second laws of thermodynamics. Numerical examples show that our novel parameterization is able to provide non-decreasing entropy that translates to improved robustness for out-of-distribution forecasting. We additionally show that exact treatment of dissipative processes allows introduction of thermal forcing

which satisfies a discrete FDT. To achieve thermodynamically consistent equilibrium distributions in this setting, we have shown the degeneracy condition must be imposed exactly.

While this work establishes the value of imposing bracket structure for dissipative processes in terms of generalization, robustness, and physical realizability, the training approach applied here is applicable only to relatively small systems, restricting its application e.g. to learning dynamics of reduced-order models. Future work will focus on developing more scalable training strategies for learning ODEs of many variables.

# 7 Acknowledgements

Sandia National Laboratories is a multimission laboratory managed and operated by National Technology and Engineering Solutions of Sandia, LLC, a wholly owned subsidiary of Honeywell International, Inc., for the U.S. Department of Energy's National Nuclear Security Administration under contract DE-NA0003530. This paper describes objective technical results and analysis. Any subjective views or opinions that might be expressed in the paper do not necessarily represent the views of the U.S. Department of Energy or the United States Government. SAND Number: SAND2021-7461 O.

We thank Chris Eldred for his recommendation to consider Ottingers bracket formulation of GENERIC. The work of N. Trask, and P. Stinis is supported by the U.S. Department of Energy, Office of Advanced Scientific Computing Research under the Collaboratory on Mathematics and Physics-Informed Learning Machines for Multiscale and Multiphysics Problems (PhILMs) project. N. Trask and K. Lee are supported by the Department of Energy early career program.

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
