$$\{\boldsymbol{x}, S\} = \frac{\partial \boldsymbol{x}}{\partial \boldsymbol{x}}L\frac{\partial S}{\partial \boldsymbol{x}} = \xi_{\alpha\beta\gamma}\frac{\partial \boldsymbol{x}}{\partial x_\alpha}\frac{\partial S}{\partial x_\beta}\frac{\partial S}{\partial x_\gamma} = \xi_{\alpha\beta\gamma}\frac{\partial S}{\partial x_\beta}\frac{\partial S}{\partial x_\gamma} = 0,$$

and

$$[\boldsymbol{x}, E] = \frac{\partial \boldsymbol{x}}{\partial \boldsymbol{x}}M\frac{\partial E}{\partial \boldsymbol{x}} = \zeta_{\alpha\beta,\mu\nu}\frac{\partial \boldsymbol{x}}{\partial x_\alpha}\frac{\partial E}{\partial x_\beta}\frac{\partial E}{\partial x_\mu}\frac{\partial E}{\partial x_\nu} = \zeta_{\alpha\beta,\mu\nu}\frac{\partial E}{\partial x_\beta}\frac{\partial E}{\partial x_\mu}\frac{\partial E}{\partial x_\nu} = 0.$$

## 4.4 Alternative parameterization – penalty-based method

An alternative strategy to incorporate GENERIC structure is to enforce the degeneracy condition by soft penalty as advocated in [46]. In this approach, $E$, $S$, $L$, and $M$, may be approximated independently of each other. Again, $E$ and $S$ are parameterized as neural networks ($E_\phi$ and $S_\varphi$), and $L$ and $M$ are parameterized as skew-symmetrizations/symmetrizations of matrices $\boldsymbol{\pi}$ and $\boldsymbol{\rho}$ with learnable entries as follows

$$L_{\boldsymbol{\pi}} = \frac{1}{2}\left(\boldsymbol{\pi} - \boldsymbol{\pi}^{\mathsf{T}}\right) \quad \text{and} \quad M_{\boldsymbol{\rho}} = \boldsymbol{\rho}\boldsymbol{\rho}^{\mathsf{T}}.$$

With this parameterization, the degeneracy conditions are simply enforced by minimizing two penalty terms, $\left\|L_{\boldsymbol{\pi}}\frac{\partial E_\phi}{\partial \boldsymbol{x}}\right\|$ and $\left\|M_{\boldsymbol{\rho}}\frac{\partial S_\varphi}{\partial \boldsymbol{x}}\right\|$. We stress that this penalty will be enforced only to within optimization error.

If we write a system of neural ODEs as $\frac{\partial \boldsymbol{x}}{\partial t} = \boldsymbol{f}_\Theta$, where $\Theta$ consists of learnable parameters, then Table 1 summarizes the components comprising $\boldsymbol{f}_\Theta$ for black-box NODE, the penalty-base method, and GENERIC NODE (GNODE).

Table 1: Model summary

| | NODE | Penalty | GNODE |
|---|---|---|---|
| $\boldsymbol{f}_\Theta$ | $\boldsymbol{f}_\Theta = \boldsymbol{f}_\theta$ | $\boldsymbol{f}_\Theta = L_{\boldsymbol{\pi}}\frac{\partial E_\phi}{\partial \boldsymbol{x}} + M_{\boldsymbol{\rho}}\frac{\partial S_{\boldsymbol{\varphi}}}{\partial \boldsymbol{x}}$ | $\boldsymbol{f}_\Theta = \{\boldsymbol{x}, E\} + [\boldsymbol{x}, S]$ |
| Components | black-box MLP | $E_\phi$ and $S_{\boldsymbol{\varphi}}$ (MLPs) $L_{\boldsymbol{\pi}}$ and $M_{\boldsymbol{\rho}}$ (2-tensor) | $E_\phi$ and $S_{\boldsymbol{\varphi}}$ (MLPs) $\xi$ (3-tensor), $\Lambda$ and $D$ (2-tensor) |
| $\Theta$ | $\Theta = \theta$ | $\Theta = \{\phi, \boldsymbol{\varphi}, \boldsymbol{\pi}, \boldsymbol{\rho}\}$ | $\Theta = \{\phi, \boldsymbol{\varphi}, \xi, \Lambda, D\}$ |

## 5 Experiments

In this section, we assess the performance of the three parameterizations of the ODE dynamics which apply progressively more stringent priors. We implement the algorithms in PYTHON 3.6.5, NUMPY 1.16.2, and PYTORCH 1.7.1 [48]. For the time integrator, we use a PYTORCH implementation of differentiable ODE solvers, TorchDiffEq [3]. All experiments are performed on MACBOOK PRO with 2.9 GHz i9 CPU and 32 GB memory.

### 5.1 Dataset and training

The states $\boldsymbol{x}$ of GENERIC systems may generally be partitioned between "observable" states (e.g., position and momentum variables) denoted by $\boldsymbol{x}^{\mathrm{o}}$ and "non-observable" states (e.g., entropy, configuration variables, etc) denoted by $\boldsymbol{x}^{\mathrm{u}}$, i.e., $\boldsymbol{x} = [\boldsymbol{x}^{\mathrm{o}\mathsf{T}}, \boldsymbol{x}^{\mathrm{u}\mathsf{T}}]^\mathsf{T}$. We assume that training data is only available for the observable states, with the non-observable states functioning as hidden variables during training. For each benchmark problem, we take as manufactured training data a single trajectory of observable states obtained by integrating a reference ODE with known GENERIC structure from a known initial condition. We then split the sequence into three segments, $[0, t_{\mathrm{train}}]$, $(t_{\mathrm{train}}, t_{\mathrm{val}}]$, and $(t_{\mathrm{val}}, t_{\mathrm{test}}]$ for training, validation, and test such that $0 < t_{\mathrm{train}} < t_{\mathrm{val}} < t_{\mathrm{test}}$.

We employ mini-batching to train all three considered architectures. Each mini-batch consists of multiple short sequences of length $L$ whose initial conditions are randomly chosen from $[0, t_{\mathrm{train}}]$. To train "black-box" neural ODEs, we simply use a stochastic gradient descent (SGD) optimizer to update the network weights and biases using the mini-batches on the observable states, $\{\boldsymbol{x}_\ell^{\mathrm{o}}, \boldsymbol{x}_{\ell+1}^{\mathrm{o}}, \ldots, \boldsymbol{x}_{\ell+L-1}^{\mathrm{o}}\}$.

As opposed to the black-box neural ODEs, training the penalty-based approach and the GENERIC approach requires data to impose mini-batch initial conditions on non-observable states, i.e., $\{\boldsymbol{x}_\ell, \boldsymbol{x}_{\ell+1}, \ldots, \boldsymbol{x}_{\ell+L-1}\}$ with $\boldsymbol{x}_\ell = [\boldsymbol{x}_\ell^{\mathrm{o}}, \boldsymbol{x}_\ell^{\mathrm{u}}]^\mathsf{T}$, where $\{\boldsymbol{x}_\ell^{\mathrm{u}}\}$ are unavailable. To address this issue, we propose a training strategy that alternately updates the model parameters and infers the non-observable states. We start with a guess for the non-observable states. We then alternate between (1) updating the model parameters using SGD while fixing the current non-observable states and (2) updating the non-observable states by solving an initial value problem using the most recent model.

---

**Algorithm 1:** Neural ODE training

1   Initialize $\Theta$
2   **for** $(i = 0;\ i < n_{\max};\ i = i + 1)$ **do**
3      Sample initial points $\{\boldsymbol{x}_{\ell(k)}^{\mathrm{o}}\}_{k=1}^{N_{\mathrm{b}}}$, where $\ell(k) \in [0, t_{\mathrm{train}} - L - 1]$ for $k = 1, \ldots, N_{\mathrm{b}}$
4      $\tilde{\boldsymbol{x}}_{\ell(k)+1}^{\mathrm{o}}, \ldots, \tilde{\boldsymbol{x}}_{\ell(k)+L}^{\mathrm{o}} = \mathrm{ODESolve}(\boldsymbol{x}_{\ell(k)}^{\mathrm{o}}, \

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

# A Additional experiments

## A.1 Damped nonlinear oscillator with the canonical Poisson matrix and measurements on $S$

In this experiment, we consider the same benchmark problem considered in Section 5.2, but with different assumptions: i) the Poisson matrix is known *a priori*, i.e.,

$$L = \begin{bmatrix} 0 & -1 & 0 \\ 1 & 0 & 0 \\ 0 & 0 & 0 \end{bmatrix},$$

and ii) the measurement on $S$ is available. Thus, for this experiment, we consider that $L$ and $S$ are given, and the energy function and the friction matrix are parameterized in the same way as in the main manuscript: an MLP with 1 hidden layer with 10 neurons and Tanh for parameterizing $E_\phi$, and $3 \times 3$ skew-symmetric tensor to parameterize $\Lambda$, and $3 \times 3$ tensor, $d$, to parameterize $D$, i.e., $D = dd^\mathsf{T}$. There is also a small modification in the training algorithm: because we assume that the measurement on $S$ is available, there is no update procedure for "non-observable" states. We employ the same hyperparameters for training the neural network as in the main manuscript.

Figure 4 reports the results obtained from 5 independent runs (i.e., 5 different random seeds). Again, the plots of $\frac{dS}{dt}$ shows that the GNODE approach enforces by construction $\frac{dS}{dt} \geq 0$ and learns a nearby entropy which consistently dissipates the correct amount of energy.

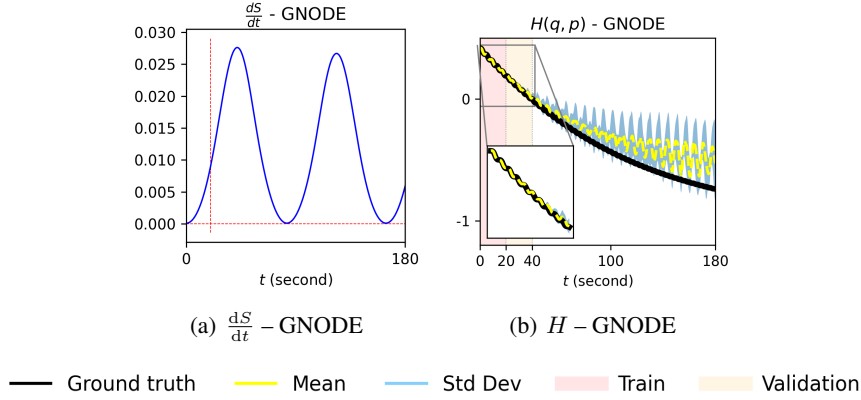

(a) $\frac{\mathrm{d}S}{\mathrm{d}t} - \text{GNODE}$   (b) $H - \text{GNODE}$

Figure 4: $\frac{dS}{dt}$ and $H(q,p)$ for damped nonlinear oscillator with known $L$ and $S$.

## A.2 Nonlinear oscillator without damping

In this experiment, we consider a non-damped-version of the benchmark problem 1 considered in Section 5.2. That is, the system of ODEs can be written as

$$\frac{\mathrm{d}q}{\mathrm{d}t} = \frac{p}{m}, \qquad \frac{\mathrm{d}p}{\mathrm{d}t} = k\sin(q), \qquad \frac{\mathrm{d}S}{\mathrm{d}t} = 0, \tag{6}$$

and the total energy simply recovers back to the Hamiltonian function,

$$E(q,p,S) = H(q,p) = \frac{p^2}{2m} - k\cos(q),$$

where $m = k = 1$.

We consider the GENERIC approach for parameterizing the dynamics function: an MLP with 1 hidden layer with 10 neurons and Tanh for parameterizing $E_\phi$, and a linear layer for parameterizing $S_\varphi$, $3 \times 3 \times 3$ skew-symmetric tensor to parameterize $\xi$, $3 \times 3$ skew-symmetric tensor to parameterize $\Lambda$, and $3 \times 3$ tensor, $d$, to parameterize $D$, i.e., $D = dd^\mathsf{T}$.

The dataset is generated by solving the initial value problem associated with Eq. (6) and the initial condition $x^0 = [1,2,0,0]$ for 180,000 timesteps with $t_{\text{final}} = 180$ (in second) and step size $\Delta t =$

0.001. Again, the dataset is split into training, validation, and testing sets such that $t_{\text{train}} = 20$, $t_{\text{val}} = 40$, and $t_{\text{test}} = 180$. Each mini-batch consists of $N_{\text{b}} = 20$ subsequences of length $L = 120$. The maximum training step is set as $n_{\text{max}} = 30000$ and the update is performed at every $n_{\text{update}} = 500$ training steps (in Algorithm 2).

Figure 5 reports the experimental results obtained from 5 independent runs (i.e.,5 different random seeds). The conservation of the Hamiltonian is enforced while maintaining $\frac{\text{d}S}{\text{d}t} > 0$.

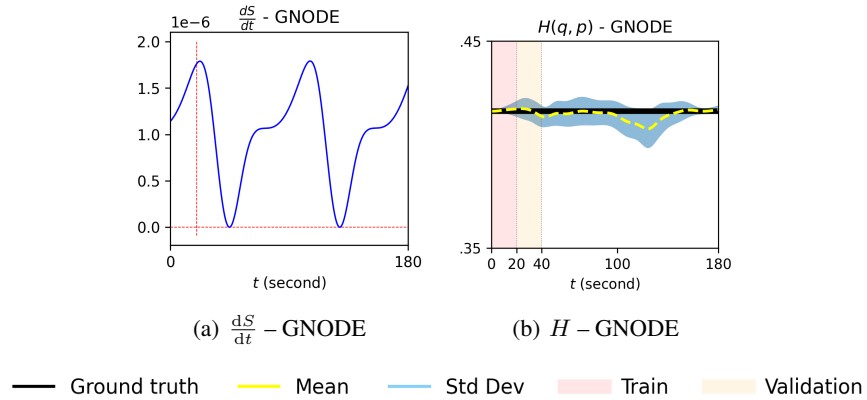

(a) $\frac{\text{d}S}{\text{d}t} - \text{GNODE}$                (b) $H - \text{GNODE}$

— Ground truth    — Mean    — Std Dev    ▮ Train    ▮ Validation

Figure 5: $\frac{dS}{dt}$ and $H(q, p)$ for non-damped nonlinear oscillator.

The Poisson matrix $L(\boldsymbol{x})$ and the friction matrix $M(\boldsymbol{x})$ can be assembled by contracting the tensors:

$$[L]_{\alpha\beta} = \xi_{\alpha\beta\gamma} \frac{\partial S}{\partial x_\gamma}, \qquad [M]_{\alpha\mu} = \zeta_{\alpha\beta,\mu\nu} \frac{\partial E}{\partial x_\beta} \frac{\partial E}{\partial x_\nu}.$$

When the matrices are evaluated at the number of sampled coordinates $\boldsymbol{x}$, the diagonal entries of $L$ is essentially zero (around machine epsilon) and the entries on the third column and the third row are three to four orders of magnitude smaller than $[L]_{12}$ ((1,2)-entry) and $[L]_{21}$ ((2,1)-entry), where the values of $[L]_{12} = -[L]_{21} = 4.5475$. All the entries of the friction matrix have the magnitudes around $10^{-8} \sim 10^{-5}$.

## B   Two gas container for varying initial conditions

Lastly, we consider a scenario where we can generate multiple trajectories for varying initial conditions to construct training/validation/test datasets. We randomly chosen initial conditions for the canonical coordinates $q(t_0), p(t_0) \in [0.5, 1.5]$ and fixed initial conditions for the entropies $S_1(t_0) = S_2(t_0) = 1.5\log(2.0) + \hat{c}$, where the $\hat{c}$ is a constant introduced in [51] and has the value 102.2476, and generate 320 training, 64 validation, and 64 test sets. All other problem parameters for characterizing the problem is the same as the one considered in Section 5.3. For generating the datasets, 5,120 timesteps with $t_{\text{final}} = 5.120$ seconds with step size $\Delta t = 0.001$ are considered and the 4th-order Runge–Kutta is used.

We consider GNODE consisting of two MLPs with 4 hidden layers and 15 neurons in each layer for parameterizing $E_\phi$ and $S_\varphi$, and consider $3 \times 3 \times 3$ skew-symmetric tensor to parameterize $\xi$, $4 \times 4$ skew-symmetric tensor to parameterize $\Lambda$, and $1 \times 1$ tensor, $d$, to parameterize $D$, i.e., $D = dd^{\mathsf{T}}$. For initializing layers in MLPs, we use the PYTORCH default uniform distribution and, for initializing learnable entries, we initialize them with unit normal distribution. Again, the non-observable variables are initialized as $s_{1,\ell} = s_{2,\ell} = t_\ell$. As the length of sequences in the datasets is relatively short (5,120 time steps), in the forward pass, the initial value problem is solved for the entire sequence (i.e., 5,120 time steps).

Figure 6 reports the results obtained from 3 independent runs (i.e., 3 different random seeds): trajectories of $q$ and $p$ obtained by solving an initial value problem with the learned dynamics function and an unseen initial condition sampled from the test set. The computed trajectories match

well with the ground truth trajectory both for $q$ and $p$ with the small values of the standard deviation. Finally, the plot of $\frac{\mathrm{d}S}{\mathrm{d}t}$ shows again that the GNODE approach enforces by construction $\frac{\mathrm{d}S}{\mathrm{d}t} \geq 0$.

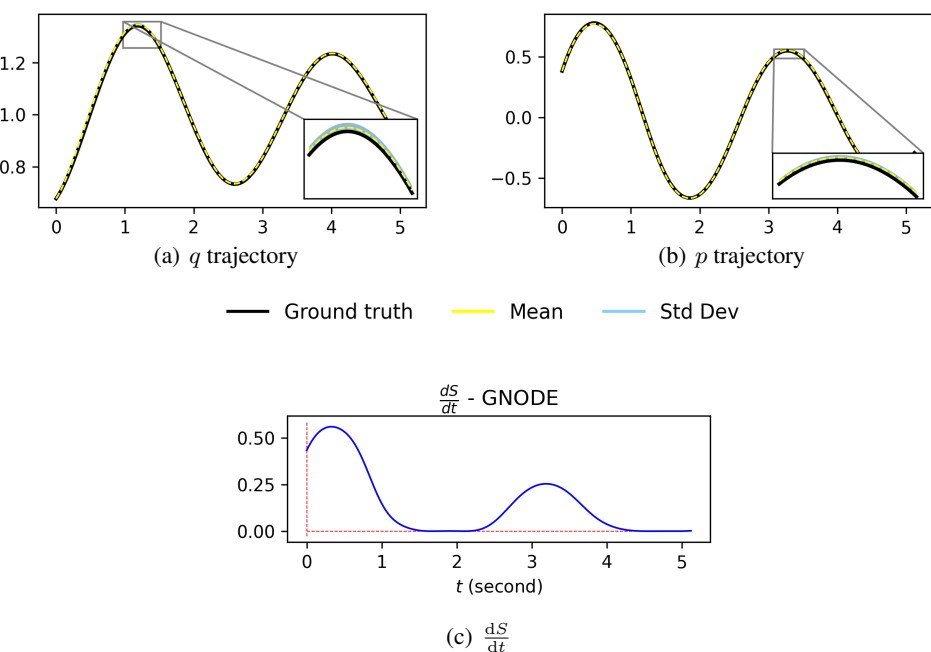

Figure 6: Trajectories of $(q,p)$ and $\frac{\mathrm{d}S}{\mathrm{d}t}$ for one test instance.