# OpenReview forum: "Machine learning structure preserving brackets for forecasting irreversible processes"
_NeurIPS.cc/2021/Conference — NeurIPS 2021 Poster_

### Official Review · Reviewer_xSwS · 2021-07-13

**Rating:** 3
**Confidence:** 3

**Summary:**

The paper proposes a new ML model parametrization for imposing irreversibility conditions on a learned physical system. The author then compares their approach to a black-box learner and an ML model where the irreversibility is indirectly added via a regularization term during training.

**Limitations And Societal Impact:**

Missing comprising to alternative extrapolation and physics-informed ML models, combined with low-dimensional datasets, makes the paper fail to show its limitations.

**Main Review:**

I believe the paper has some merits. However, it also faces serval weaknesses that need to be addressed.

**Writing**: From reading the paper, it is very evident that the authors do not come from ML but a physics background (e.g., use of "penalty" instead of "regularization", algorithm 1 is trivial, etc.). As a result, this makes the paper harder to read for the NeurIPS community. For instance, in section 4.1, the paper starts using $x_\alpha$, $x_\beta$, $x_\gamma$, and $x_\nu$ without defining them explicitly. Thus it is unclear how the final GNODE model is structured. Moreover, the bracket structures are defined twice, and it is not clear if one is a special case of the other.

The experiment part has three issues:

**Few experiments**: I don't think one can make general claims from having only two experiment datasets (plus a stochastic version of one). Especially as the datasets are pretty toyish, i.e., the two datasets are relatively low dimensional. It is well established in ML that many models scale poorly with dimension (i.e., the curse of dimensionality). Therefore, different and higher dimensional datasets would enormously benefit the paper's claims.

**Lack of quantitative comparisons**. The gas container results have no mean and standard deviation, i.e., how does the average error look over serval repeated training runs. Connecting to the issue above: More datasets are necessary to support the general claims.

**Missing comparison with alternative approaches**. There is some substantial literature on incorporating physics-based priors to ML models, for instance [1]. Furthermore, the paper investigates extrapolation with ML models. This is a sub-field of ML where it is well known that existing methods yield non-optimal performance, and more task-specific models work much better [2,3]. Therefore, the paper should experimentally evaluate how these models compare to the proposed approach.

In conclusion, as long as the paper lacks more complex datasets and comparisons with more diverse existing approaches, I firmly believe this paper does not suffice the standard of this conference, and I favor rejection. I urge the authors to either resubmit the paper to a more suited computational physics venue or substantially scale up their experimental evaluation and clarify the writing.

[1] Patel et al. A physics-informed operator regression framework for extracting data-driven continuum models. 2020
[2] Sahoo et al. Learning Equations for Extrapolation and Control. ICML 2018
[3] Madsen and Johansen. Neural Arithmetic Units. ICLR 2020

**Time Spent Reviewing:**

4

---

> ### Author Response · Authors · 2021-08-10
> **Response to Reviewer xSwS**
>
> We thank the reviewer for taking the time to read the paper and provide valuable feedback. We respond to points inline below - responses to quoted feedback are prefaced with (--).
>
> - "Missing comparison with alternative approaches. There is some substantial literature on incorporating physics-based priors to ML models, for instance [1]. Furthermore, the paper investigates extrapolation with ML models. This is a sub-field of ML where it is well known that existing methods yield non-optimal performance, and more task-specific models work much better [2,3]. Therefore, the paper should experimentally evaluate how these models compare to the proposed approach."
> --- We are deeply familiar with the cited references, which similarly aim toward structure preservation but are in our opinion not suitable for comparison to the problem at hand. We are interested in learning thermodynamically consistent dynamics $\dot{x} = F(x;\theta)$ where the parameterization has attractive "black-box" universal approximation properties and does not require assuming a dictionary.
> ---Reference [1] aims to infer a partial differential equation and considers inductive priors tailored toward conservation law structure, which is an important but completely separate type of structure preservation. To be concrete, discrete preservation of the 1st and 2nd law of thermodynamics implies that a data-driven model is rigorously guaranteed to be numerically stable. Conservation of flux does not give this robustness guarantee. Reference [2] aims to use dense networks to construct a dictionary of models similar to Kutz+Brunton's SINDy work (Kaiser, E., Kutz, J.N. and Brunton, S.L., 2018. Sparse identification of nonlinear dynamics for model predictive control in the low-data limit. Proceedings of the Royal Society A, 474(2219), p.20180335.). While this is able to build complex dictionaries that provide much needed interpretability, it (1) requires specification of atomistic dictionary components in the layer neurons and (2) has an explosion of required network size as the dimension increases and (3) provides no notion of structure preservation. Reference [3] (similar to [2]) aims to learn interpretable formulas, but not in the context of dynamical systems.
> --- From our perspective and to the best of our knowledge, NODE and the penalty formulation of GENERIC were the only applicable techniques in the literature as candidates for comparison. As noted in the paper, this specific type of structure preservation has seen an explosion of papers in the past two years at neurips/ICLR/ICML/etc (see e.g. refs 4-10 in paper). However, these papers which focus exclusively upon *reversible* systems cannot be fairly compared to, since they by construction cannot model dissipation and will give a poor prediction. For this reason the only "black-box" technique we could compare to was NODE (although one could arguably consider similar architectures like resnets/RNNs). Our focus however was to propose, analyze, and validate a new architecture rather than provide a benchmark across existing methods, and due to space constraints felt it better to choose one representative "black-box" approach and spend more space articulating technical details of the architecture.
> --- We next clarify why this "black-box" property is so important. It may not be obvious at a first pass, but the two gas container problem is deceptively challenging and in our experience forces dictionary-based learning strategies to fail. The reason for this is that the expression for the entropy s_i cannot be easily expressed in terms of linear combinations of dictionary terms; the term is the log of a constant of a variable and therefore the parameters which must be learned are "inside" the log.  The only way to obtain this model is to use a dictionary which includes the dynamics that we are attempting to learn - however if one were able to do this then there would be no need to machine learn a forcing term. For realistic problems of interest to physical sciences, and in particular those related to coarse-grained/reduced order models of complex systems, this is most often the case.
>
> - "Lack of quantitative comparisons. The gas container results have no mean and standard deviation, i.e., how does the average error look over serval repeated training runs. Connecting to the issue above: More datasets are necessary to support the general claims."
> --- As mentioned in our response to other reviewers comments, at the time of writing we were attempting to learn the dynamics from a single trajectory. With this setup, the training was sensitive to the random initialization. We have since slightly altered the training strategy to allow batching over trajectories corresponding to several different initial conditions which trains much better and we will include these statistics (i.e. mean+standard deviation) in the revised manuscript.
> --- We note that our intent with these examples was to validate that structure preservation held and gave improved prediction for long term integration. As indicated in the manuscript, there is room for improvement regarding training, as we simply applied Adam to the architecture with very little tuning.
>
> - "Writing: From reading the paper, it is very evident that the authors do not come from ML but a physics background (e.g., use of "penalty" instead of "regularization", algorithm 1 is trivial, etc.). As a result, this makes the paper harder to read for the NeurIPS community. For instance, in section 4.1, the paper starts using , , , and  without defining them explicitly. Thus it is unclear how the final GNODE model is structured. Moreover, the bracket structures are defined twice, and it is not clear if one is a special case of the other."
> --- Thank you for this suggestion. The notation used is the well-known Einstein/multi-index notation (https://en.wikipedia.org/wiki/Einstein_notation). We've added a comment to clarify this in the text. From the feedback from the other reviewers it seems there is a consensus that the paper is well written, and so we are assuming that the use of standard mathematical phrases is acceptable for neurips.

---

> > ### Comment · Reviewer_xSwS · 2021-08-22
> > **Response to the authors**
> >
> > I thank the authors for their rebuttal. However, I am not really satisfied with the response.
> >
> > *Writing* None of my issues about the writing concerns the Einsum notation, which the author's response exclusively focuses on. Thus my issues regarding the writing are not addressed at all.
> >
> > *Alternative approaches* Neural ODEs are defined with respect to arbitrary neural network dynamic functions. Therefore, any method developed for static extrapolation can be incorporated into a NODE setting. Besides, all deep learning frameworks (PyTorch, TensorFlow, JAX) have an ODE solver implementation. So there is no reason why ML extrapolation methods are inapplicable in the paper's context.
> >
> > *black-box property* The arguments about the learnable components inside the logarithm are precisely that: arguments.
> > In ML, all models are wrong, but some are useful. As this paper has been submitted to an ML venue, I have to evaluate it from an ML perspective. For a counterargument, one could simply approximate the log term arbitrarily close or have a stable network without a formal stability proof. I strongly believe the authors need to evaluate whether this "approximation" fails in existing ML models and that the authors' parametrization can scale to learning complex dynamics (i.e., does not prevent learning). As mentioned in my review, the paper falls short on this, as 1) from the class of existing ML models, only a Neural ODE is used as the baseline. As the tasks in the experiment concern time-series data of dynamical systems, any recurrent neural network could be applied here.
> > 2) The amount of learning tasks is small. Moreover, the presented datasets are all low dimensional, as I mentioned.
> > If authors argue that the presented datasets are difficult, then show it by evaluating how other types of existing ML models fail here (as mentioned above).
> >
> > In conclusion, as I mentioned in the review, I think this paper has potential, and I have no criticisms of the proposed parameterization.
> > However, the paper should involve less arguing and more empirical evidence to convey its significance for the ML domain.

---

### Official Review · Reviewer_DeZu · 2021-07-14

**Rating:** 6
**Confidence:** 4

**Summary:**

The paper introduces a framework for learning dissipative dynamics efficiently by imposing a physical bias in the network architecture based on the GENERIC formalism. This framework is  commonly seen in the nonequilibrium statistical mechanics literature, where one models the conservative dynamics using a Poisson bracket and the dissipative dynamics using a metric bracket. However, in contrast to what is usually done in physics, where such algebraic structures are observed after deducing the physical laws from first principles, the authors start by assuming such structure in the data and learn dynamics that conform to it. Experiments are done on simple physical systems demonstrating that this approach successfully captures correct physical behaviours (such as the second law of thermodynamics) and has better long term predictions as compared to a neural ODE baseline and another baseline that enforces a weak constraint on the GENERIC structure through a penalty term.

**Limitations And Societal Impact:**

- As the authors point out, one of the limitations of this approach is scalability to higher dimensions. A further discussion about this is certainly desired. What prevents it from scaling to high dimensions? Also how much additional speed and memory cost does this method add to training a vanilla neural ODE?
- Discussion of how this model can be applied in real world settings is lacking.


**Main Review:**

Originality:
I believe that the approach presented in this paper, while clearly following the footsteps of recent works on imposing physical constraints in neural networks, has not been done before. In particular, the explicit parameterisations of the conservative and dissipative brackets presented in Sections 4.1 and 4.2 are appealing as it makes it easy to integrate with Neural ODE packages as opposed to other structure-preserving methods based on geometric integrators, and enables a better enforcement of the GENERIC conditions as opposed to the approach using penalty terms [41].

Quality:
Overall the paper is of good quality that reads nicely and easy to follow, except for a few places that I note below. It would have also been nice to see some experiments using real data to demonstrate the ubiquity of this modelling assumption. The experiments here use only toy problems that is exactly a GENERIC system, so it is somewhat expected that the method works well. The experiment in section 5.4 is also not very clear to me as I explain in more details below.

Clarity:
- Lines 101-102. When stating that the explicit parameterisations of the brackets allows an exact enforcement of the GENERIC conditions, I understand that this holds theoretically, but is this still satisfied under the dopri5 numerical discretisation?
- Line 160. The following line I think needs clarification: "As opposed to the black-box neural ODEs, training the penalty-based approach and the GENERIC approach requires data to impose mini-batch initial conditions on non-observable states". This is something that might be obvious when implementing the algorithm, but I think it is not so clear from what is discussed in the preceding paragraphs.
- Lines 181-186: The lowercase $s$'s are perhaps meant to be uppecase $S$'s? Same for lines 224-229.
- Equation below line 216: Please check again. I think one of the $S_1$'s should be $S_2$.
- Figures 1 and 2: What do the dashed red lines in the top left and centre images indicate? If it is important, it should perhaps be made bolder.
- Figure 2. The dotted yellow line makes the figure a little hard to see. Perhaps it's better to make it a solid line with some opacity?
- I find that the experimental setup in section 5.4 is not clearly written. I am confused as to what is being learnt here: is the data now generated by the SDE? Or are you fitting the SDE on the same data as 5.2? If fitting an SDE, details are necessary as this is not straightforward.
- Line 279. A further discussion about the computational cost (speed, memory, etc) is desired, following the statement "the training approach applied here is applicable only to relatively small systems, restricting its application". Is it because of the parameterisations of the symmetric and antisymmetric brackets which contain a four-tensor and a three-tensor respectively and therefore requiring large memory? Would it help to use a GPU instead?

Other suggestions:
- I suggest replacing the word "obtain" in line 31 with "exhibit"
- Line 52. "with Hamiltonian structure particularly useful" -> "with the Hamiltonian structure being particularly useful"
- Line 81. "Consider time series data" -> "Consider the time series data"

Significance:
I believe that this work will provide a way to learn more realistic systems from data beyond simple conservative systems, but restricted enough to satisfy thermodynamic laws. Showing experiments on real data would have made the case stronger.


**Time Spent Reviewing:**

5

---

> ### Author Response · Authors · 2021-08-09
> **Response to Reviewer DeZu**
>
> We thank the reviewer for their time reviewing the paper, and positive impression of our work. In other comments we have discussed some aspects of our ongoing work scaling this to higher dimensions, and we focus in this response upon the list of concerns given. Suggestions regarding grammar+style, figure coloring, and typos with entropy subscripts (S_i) have been incorporated into the manuscript. Quotes are preceeded by (-), with responses (---)
>
> - "Lines 101-102. When stating that the explicit parameterisations of the brackets allows an exact enforcement of the GENERIC conditions, I understand that this holds theoretically, but is this still satisfied under the dopri5 numerical discretisation?"
> --- While the ODE is discretized with dopri5 for the purposes of training, inference of the learned ODE is obtained by discretizing the learned continuum ODE (i.e. dx/dt = F(x)) which may be discretized with any numerical integrator. For classical multi-stage solvers such as Runge-Kutta, this means that the GENERIC conditions will be realized in the limit as timestep DT approaches zero. A more sophisticated treatment (Öttinger, Hans Christian. "GENERIC integrators: structure preserving time integration for thermodynamic systems." Journal of Non-Equilibrium Thermodynamics 43, no. 2 (2018): 89-100.) generalizing exponential integrators guarantees that structural guarantees (e.g. dE/dt = 0, dS/dt \geq 0 ) hold even for finite timesteps. We stress that this only affects *inference*, as at the continuum level the extracted ODE satisfies GENERIC structure by construction independent of the timestep size. We've added a comment to this effect to our manuscript.
>
> - "Line 160. The following line I think needs clarification: "As opposed to the black-box neural ODEs, training the penalty-based approach and the GENERIC approach requires data to impose mini-batch initial conditions on non-observable states". This is something that might be obvious when implementing the algorithm, but I think it is not so clear from what is discussed in the preceding paragraphs."
> --- We have revised this statement. The difference is that the black-box neural ODE does not aim to discover a hidden state and only aims to solve an ODE for the observed quantities (in these examples, an ode for position and momentum only). Therefore, minibatches may use the data for position and momentum as an initial condition. For the GENERIC models an internal state variable it required - for example the entropy s_i may be thought of as modeling the internal configuration which stores energy internally, such as how a temperature characterizes the vibrational energy of a system. As these variables can't be observed experimentally, it is non-physical to assume we have access to data for them which may be used as an initial condition for a minibatch.
>
> - "I find that the experimental setup in section 5.4 is not clearly written. I am confused as to what is being learnt here: is the data now generated by the SDE? Or are you fitting the SDE on the same data as 5.2? If fitting an SDE, details are necessary as this is not straightforward."
> --- For this section, we fit a deterministic model to the deterministic data, which provides an estimate for L, M, E, and S. We contrast this against the true models L_T, M_T, E_T, and S_T. Following the fluctuation dissipation theorem, we then define two SDEs
> dx  = L grad(E) + M grad(S)dt + \sqrt{2 M} dB    and     dx_T  = L_T grad(E_T) + M_T grad(S_T)dt + \sqrt{2 M_T} dB
> and use Euler-Maruyama to solve both and compare the resulting distribution at the final time. This confirms that the learned dynamics provide converged equilibrium statistics and that the approach is suitable for modeling thermal systems. It may not be obvious, but models which do not satisfy this fluctuation-dissipation theorem do not give converged equilibrium statistics. We thank the reviewer for pointing out that this section could use revision, and have revised our description for the final manuscript.
>
> - Line 279. A further discussion about the computational cost (speed, memory, etc) is desired, following the statement "the training approach applied here is applicable only to relatively small systems, restricting its application". Is it because of the parameterisations of the symmetric and antisymmetric brackets which contain a four-tensor and a three-tensor respectively and therefore requiring large memory? Would it help to use a GPU instead?
> --- For this problem, the necessary variables are the tensors for M and L, along with two neural networks for E and S. For the tensors M and L, they are (as presented) dense and therefore scale with complexity O(N^3) and O(N^4) for the 3- and 4- tensors in M and L, respectively. For practical applications, there are typically problem specific sparsifications of these tensors available - for example in particle systems one may define finite interactions between neighbors to obtain sparse tensors. In addition to this memory consideration, since submitting the draft we have developed an effective training strategy by batching over several initial conditions concurrently rather than learning dynamics from a single trajectory, which has allowed us to consider larger systems. We intend to introduce these results into the final draft of the paper.

---

> > ### Comment · Reviewer_DeZu · 2021-08-24
> > **Rebuttal response**
> >
> > I thank the authors for the detailed response. Upon further examination and looking through the other reviews, I regret to say that I have lowered my score from 7 to 6 for now, but willing to push back to 7 again if the following points are addressed adequately. This mainly stems from the weaknesses in the experiments section as other reviewers have noted (setup is too simple, scalability issues, not enough baselines). However, I do maintain my enthusiasm for this work as I think the idea of exploiting the GENERIC inductive bias to model irreversible systems is promising and has not been explored enough in the literature, which often focuses too much on modelling reversible dynamics. Therefore I wholeheartedly support the work, however there are some things that should be addressed.
> >
> > > We have revised this statement. The difference is that the black-box neural ODE does not aim to discover a hidden state and only aims to solve an ODE for the observed quantities (in these examples, an ode for position and momentum only). Therefore, minibatches may use the data for position and momentum as an initial condition. For the GENERIC models an internal state variable it required - for example the entropy s_i may be thought of as modeling the internal configuration which stores energy internally, such as how a temperature characterizes the vibrational energy of a system. As these variables can't be observed experimentally, it is non-physical to assume we have access to data for them which may be used as an initial condition for a minibatch.
> >
> > This makes sense. Thanks for clarifying. However, I think that it may be more fair to compare with latent neural ODEs, where the ODE dynamics take place in a latent space that is different from the observation space (i.e. encode first into latent space, progress dynamics there, then decode). In this setting, the neural ODE might actually be able to implicitly learn the effect of the entropy on the position and momentum, giving results that are harder to beat. What do the authors think about this?
> >
> > > We contrast this against the true models $L_T$, $M_T$, $E_T$, and $S_T$. Following the fluctuation dissipation theorem, we then define two SDEs $dx = L \nabla E + M \nabla S dt + \sqrt{2 M} dB$ and $dx_T = L_T \nabla E_T + M_T \nabla S_Tdt + \sqrt{2 M_T} dB$ and use Euler-Maruyama to solve both and compare the resulting distribution at the final time. This confirms that the learned dynamics provide converged equilibrium statistics and that the approach is suitable for modeling thermal systems.
> >
> > Understood. So the aim here is to verify that the learned $L, M, E$ and $S$ satisfy the correct fluctuation-dissipation relation. However,
> >
> > > For this section, we fit a deterministic model to the deterministic data
> >
> > is it a realistic assumption that the observed data comes from a deterministic system? If one were to model a thermal system, I would have imagined the data would come from a thermal system too, which is stochastic. How does one deal with this stochasticity when trying to learn the parameters of the model?
> >
> > >  For practical applications, there are typically problem specific sparsifications of these tensors available - for example in particle systems one may define finite interactions between neighbors to obtain sparse tensors. In addition to this memory consideration, since submitting the draft we have developed an effective training strategy by batching over several initial conditions concurrently rather than learning dynamics from a single trajectory, which has allowed us to consider larger systems.
> >
> > Please could you elaborate further on this? Since the $O(N^3)$ and $O(N^4)$ cost for the $M$ and $L$ tensors respectively is pretty big, I think this should be addressed properly in order to make the algorithm practically useful. In particular, how do we exploit the finite interaction range to reduce the number of trainable parameters? What are the details of the new experiments/training procedure and how large can you it scale up?

---

### Official Review · Reviewer_AB4a · 2021-07-16

**Rating:** 5
**Confidence:** 4

**Summary:**

This paper considers the learning and forecasting of structured dynamics, in particular Poisson mechanical systems (which is a generalization of canonical Hamiltonian systems) with additional dissipation. An existing mathematical tool called the GENERIC framework is employed to represent the latent structured dynamics, and ingredients of this representation are approximated by neural networks. Some numerical experiments are provided.

**Main Review:**

The idea of this paper is very interesting. The learning and forecasting of conservative/reversible mechanical dynamics has been an emerging but very active research area, and I can see methods for learning and forecasting dissipative mechanical dynamics corresponding to important applications. Nevertheless, the latter has not been sufficiently considered by the current research. This work nicely combines nontrivial existing tools in a novel way and provides a convincing approach. The paper is also well written and well organized.

My main questions centered around the degeneracy condition (2). Could the authors comment on how restrictive it is? Would it be harder or easier to satisfy in higher dimensions? All the numerical experiments are 1 degree-of-freedom (i.e., 2 dimensional), and more complex examples could make the paper even more convincing.

In addition, the following could reduce bias in the list of existing "structure-reserving models of reversible dynamics" and make it more representative: [Bertalan et al. On learning Hamiltonian systems from data. Chaos 2019], [Zhong et al. Symplectic ODE-net. ICLR 2020], [Chen & Tao. Data-driven Prediction of General Hamiltonian Dynamics via Learning Exactly-Symplectic Maps. ICML 2021]. The 2nd one is related to NODE. The 3rd one gives a strong support of the authors’ statement that symplecticity improves the accuracy of forecast; the currently cited reference [9] for supporting this statement is a great and very important paper, but it is not about forecast. In addition, [21] is not a preprint and it is the same as [36], which appeared in ICLR 201*9*.

I’m willing to change my score after the authors’ responses, which I look forward to.


**Time Spent Reviewing:**

3.5

---

> ### Author Response · Authors · 2021-08-09
> **Repose to reviewer Reviewer AB4a**
>
> We thank the reviewer for their kind and thoughtful suggestions. We have added all suggested references to our working draft which will be updated in the final submission, along with minor comments. We address major comments point-by-point below:
>
> - "My main questions centered around the degeneracy condition (2). Could the authors comment on how restrictive it is? Would it be harder or easier to satisfy in higher dimensions? All the numerical experiments are 1 degree-of-freedom (i.e., 2 dimensional), and more complex examples could make the paper even more convincing."
> --- The treatment of the degeneracy condition is general - a broad range of physical systems may be characterized in this formalism. See for example the nice review (Öttinger, H. C. (2018). GENERIC: Review of successful applications and a challenge for the future. arXiv preprint arXiv:1810.08470.) along with the 950+ references for the textbook (Öttinger, Hans Christian. Beyond equilibrium thermodynamics. John Wiley & Sons, 2005.). This includes both finite dimensional particle systems, and infinite dimensional field theories.
> --- The reviewers main point/focus however may lie in the numerical analysis of the numerically stiff optimization problem corresponding to our specific architecture, and whether it is actually possible to fit more sophisticated/higher-dimensional models with the proposed strategy. It is indeed a challenge to fit the current model using first-order optimizers. Since submitting the draft of this paper, we have extended our approach to minibatch over multiple trajectories corresponding to different initial conditions simultaneously, as opposed to the results presented, which attempt to learn the dynamics from a single trajectory. This has allowed us to move toward larger and more chaotic systems, and we have updated the results in our working manuscript to reflect this. In general, we would anticipate application to substantially large systems (e.g. 1+M DOFs) may require the use of more sophisticated optimizers involving state-of-the-art treatment - e.g. second-order pde-constrained optimizers using checkpointing, Bayesian treatment of hidden state, etc to scale up, which we are currently working on. Additionally, for specific applications there are effective tricks to reduce an N-body system to a smaller dimension. For example, for N-body dissipative particle dynamics, one typically often works with an additive decomposition E = \sum_{i,j} F(X_i,X_j), that is to say, all particles would obey the same per particle energy+entropy, so that one would learn a low-dimensional function shared across the high-dimensional system. (Espanol, P. and Revenga, M., 2003. Smoothed dissipative particle dynamics. Physical Review E, 67(2), p.026705.)
>
> - "The learning and forecasting of conservative/reversible mechanical dynamics has been an emerging but very active research area, and I can see methods for learning and forecasting dissipative mechanical dynamics corresponding to important applications. Nevertheless, the latter has not been sufficiently considered by the current research."
> --- We agree that the presented examples do not constitute a major advance, but pedagogically provide nice analytic solutions which may be used to benchmark and validate our theoretical results; our focus is on defining the GENERIC framework in the abstract. We are current applying this framework to more sophisticated problems in science and engineering, and for specific applications require considerations regarding data management and physical models which we decided may be of less interest to the neurIPs community. Besides data-driven physics modeling, existing reversible structure-preserving architectures have been shown to improve image recognition (see e.g. Chang 2017, Haber 2017) and we anticipate that the appropriate treatment of dissipation advanced in this work may improve classical ML tasks as well, therefore we have chosen to present the idea in the abstract here with the aim of reaching a larger audience.
> --- We took this position following the example of earlier works accepted to neurIPs in structure preserving discovery of dynamics. For example, ("Hamiltonian Neural Networks" Greydanus et al, NeurIPS 2019) pursued similar toy problems, and we decided pedagogically that this introduction would minimize the need to introduce application specific jargon.
> ---We will comment however that theoretical results established here allow immediate impact in a wealth of physics and social applications where dissipation effects and their consequent fluctuations are critical, particularly for statistical physics and large deviations. For example: discovery of non-equilibrium chemistry in both biological systems and chemical engineering where traditional modeling requires simplistic equilibrium assumptions (Xiong, Jie, and Tong Zhou. "A kalman-filter based approach to identification of time-varying gene regulatory networks." PloS one 8, no. 10 (2013): e74571.); large deviations analysis of cascading power grid failure where first-principle models are unknown, but are driven by fluctuations (RRoth, Jacob, David A. Barajas-Solano, Panos Stinis, Jonathan Weare, and Mihai Anitescu. "A kinetic monte carlo approach for simulating cascading transmission line failure." arXiv preprint arXiv:1912.08081 (2019).); modeling of social systems by learning population dynamics for which diffusion is known to be important (Short, Martin B., P. Jeffrey Brantingham, Andrea L. Bertozzi, and George E. Tita. "Dissipation and displacement of hotspots in reaction-diffusion models of crime." Proceedings of the National Academy of Sciences 107, no. 9 (2010): 3961-3965.); turbulence modeling where thermodynamic balances are critical to obtains correct cascades across physical scales (Sirovich, L. ed., 2012. New perspectives in turbulence. Springer Science & Business Media.). For each of these fields, the structure-preservation provided in this paper provide theoretical guarantees that data-driven models be physically realizable without the need to introduce restrictive inductive biases on training. This is of extreme importance for fields involving multiscale dynamics (fluid flow, chemistry, biomechanics, power grid), where appropriate treatment of molecular processes mandate an exact balance between dissipation and thermal fluctuations.
> ---In short, we view this paper as the foundation for advances in a diverse number of fields. In our opinion, it is necessary for machine learned surrogates to provide these kinds of rigorous guarantees commonplace in traditional modeling before they can be adapted in high consequence science and engineering problems.
>
> We thank the reviewer again - unfortunately it is difficult to give this kind of context within the page constraints of the manuscript.

---

> > ### Comment · Reviewer_AB4a · 2021-08-11
> > **Further discussion**
> >
> > Thanks for the reply. Since I gave a positive initial rating, I hope my comments could be understood as constructive ones that aim at helping further improve the paper. Perhaps I wasn’t clear (sorry) so let me further clarify.
> >
> > - Authors’ response to “…Nevertheless, the latter has not been sufficiently considered by the current research…”
> >
> > ‘Current research’ in my context meant existing published research, not the authors’ NeurIPS submission. What I wrote was to acknowledge the importance and novelty of your submission. It meant the problem you study is a very important one and yet insufficiently discussed in the existing literature. I think you misunderstood.
> >
> > - the degeneracy condition (2)
> >
> > I still would like to understand `how restrictive it is` and `would it be harder or easier to satisfy in higher dimensions`. GENERIC is an interesting formulation not entirely known to the machine learning community, and I think readers like me would like to gain scientific understanding from your paper. Referring us to the 950+ references of a textbook helps us very little.
> >
> > - my suggestion on reducing the error and bias of the list of existing work was perhaps missed, but I still think this is important.

---

> > > ### Author Response · Authors · 2021-08-11
> > > **Response 2 to AB4a**
> > >
> > > Thanks for the quick response!
> > >
> > > - Thank you for the clarification regarding current research - we wanted to make sure we were articulating how important this field of research (structure preserving dissipative systems) has become outside of ML. Regarding your second bullet, we may have misunderstood and thought you were asking about it being restrictive in the sense of the theory applying to high-dimensional systems - the list of references was meant to point toward exciting high-dimensional applications of this GENERIC theory which we aim to introduce data-driven modeling to. Each of these references is an example of a physical system which our parameterization/architecture is able to "catch".
> > > - The real challenge moving to higher dimensions (which we understand is what was meant by restrictive) is that the training becomes harder and memory intensive. As written, our formulation yields dense 3- and 4- tensors parameterizing the brackets. For O(1-10) DOFs this is not a challenge, but moving to millions and billions of DOFs will lead to parameters that scale with the 3rd and 4th powers of the DOFs. For these larger problems there are a number of strategies for obtaining sparse brackets that instead scale as O(N). There are examples of these strategies in the literature: see for example smoothed dissipative particle dynamics (Espanol, Pep, and Mariano Revenga. "Smoothed dissipative particle dynamics." Physical Review E 67.2 (2003): 026705.), which describe N-body dynamics within the GENERIC framework. For this problem, while the model describes millions/billions of particles, the GENERIC brackets only need to parameterize a single pairwise interaction shared by all particles - therefore the O(N^4) computation cost corresponds only to N = 4 and there is no challenge.
> > > - Scaling to large systems however requires application specific tricks, and so for the current manuscript we felt it best pedagogically to stick to simple and small problems to keep the presentation clean. We have added comments regarding this line of questioning to the manuscript however.

---

> > > > ### Comment · Reviewer_AB4a · 2021-08-17
> > > > **Further discussion 2**
> > > >
> > > > I thank the authors for additional replies. However, my itemized concerns and suggestions have not been addressed.
> > > >
> > > > Take my question about the degeneracy condition for instance. $L\frac{\partial S}{\partial x}=0$ is a system of $d$ PDEs that $d$-by-$d$ skew-symmetric Poisson matrix $L$ and scalar valued $S$ have to satisfy. A similar requirement is on $M$ and $E$. It is important to understand how easy it is to satisfy those, especially if the latent system is a genuinely high-dimensional one (meaning that different degrees of freedom are coupled). Let me give a specific example: the Hamiltonian dynamics in Remark 3.1 correspond to $S=0$ and $M=\textbf{0}$ (matrix), so the degeneracy condition is satisfied, but this is a trivial case for which one doesn't even need GENERIC. What if none of $S$, $E$, $M$, $L$ is zero?

---

> > > > > ### Author Response · Authors · 2021-08-17
> > > > > **Response to reviewer AB4a**
> > > > >
> > > > > Understood, thank you for the clarification. This condition is satisfied to machine precision independent of the dimension of the problem. This is discussed in section 4.3, but we include a more detailed derivation of this result below, to hopefully make it more clear how this is ensured independent of the dimension. We've also added a comment to the manuscript that makes it clear this holds independent of dimension.
> > > > >
> > > > > To see this, compute:
> > > > > $L_{\alpha \gamma} \partial_{x_\gamma} S = \xi_{\alpha \beta \gamma} \text{ } \partial_{x_\beta} S \text{ } \partial_{x_\gamma} S$
> > > > > Switching from Einstein summation:
> > > > > $ = \frac12 \left(\sum_{\beta,\gamma} \xi_{\alpha \beta \gamma} \text{ } \partial_{x_\beta} S \text{ } \partial_{x_\gamma} S + \sum_{\beta,\gamma} \xi_{\alpha \beta \gamma} \text{ } \partial_{x_\beta} S \text{ } \partial_{x_\gamma} S \right)$
> > > > > Swapping $\beta$ and $\gamma$ in the second sum:
> > > > > $ = \frac12 \left(\sum_{\beta,\gamma} \xi_{\alpha \beta \gamma} \text{ } \partial_{x_\beta} S \text{ } \partial_{x_\gamma} S + \sum_{\gamma,\beta} \xi_{\alpha \gamma \beta} \text{ } \partial_{x_\beta} S \text{ } \partial_{x_\gamma} S \right)$
> > > > > Applying skew-symmetry condition of \gamma:
> > > > > $ = \frac12 \left(\sum_{\beta,\gamma} \xi_{\alpha \beta \gamma} \text{ } \partial_{x_\beta} S \text{ } \partial_{x_\gamma} S - \sum_{\gamma,\beta} \xi_{\alpha \beta \gamma} \text{ } \partial_{x_\beta} S \text{ } \partial_{x_\gamma} S \right)=0$.
> > > > >
> > > > > This works similarly for the other degeneracy condition. As you can see in the derivation, this does not involve the dimension in any way, and relies solely upon construction of tensors $\xi$ with appropriate anti-symmetry conditions, as is handled in Sections 4.1 and 4.2. As noted in the earlier response, there is increasing computational expense regarding our construction as the dimension increases, but the condition is always satisfied exactly. The application specific tricks mentioned in the previous response detail mitigation strategies for this expense, however the degeneracy conditions always hold exactly.
> > > > >
> > > > > This is a critical component of the method. If these conditions do not hold exactly, then one does not obtain the first law of thermo (dE/dt = 0), the second law (dS/dt \geq 0) or the fluctuation dissipation theorem. This is precisely why penalty methods are ineffective for handling these types of systems as degeneracy holds only loosely, leading to predictions violating the 1st and 2nd law (see e.g. Figure 1).

---

> > > > > > ### Comment · Reviewer_AB4a · 2021-08-17
> > > > > > **My concerns were not addressed**
> > > > > >
> > > > > > This is not what I asked. Neither were my other itemized points addressed. Besides, you wrote $L_{ij}\partial_{xj}S = \xi_{\alpha\beta\gamma}\partial_{x\beta} S\partial_{x\gamma}S$, but how come it is $i$ on the left but $\alpha$ on the right? I lowered my rating from 6 to 5.

---

> > > > > > > ### Author Response · Authors · 2021-08-17
> > > > > > > **response**
> > > > > > >
> > > > > > > Apologies, the i index should have been an alpha. I've updated the comment.
> > > > > > >
> > > > > > > Can you please explain what you meant? I'm concerned we're misunderstanding your question.
> > > > > > >
> > > > > > > When you ask "It is important to understand how easy it is to satisfy those, especially if the latent system is a genuinely high-dimensional one (meaning that different degrees of freedom are coupled)". The answer to that question is that this parameterization automatically enforces those constraints to machine precision independent of the dimension. In your example (a high-dimensional system of DOFs corresponding to a PDE system), this would correspond to take X as those DOFs, and the construction would carry through automatically.

---

> > > > > > > > ### Comment · Reviewer_AB4a · 2021-08-25
> > > > > > > > **Re**
> > > > > > > >
> > > > > > > > Like previously repeated, I'd like to see:
> > > > > > > >
> > > > > > > > - More explanation of how broad the scope of GENERIC is, especially in terms of what kind of systems satisfy the degeneracy condition.
> > > > > > > > "Take my question about the degeneracy condition for instance. $L\frac{\partial S}{\partial x}=0$ is a system of $d$ PDEs that $d$-by-$d$ skew-symmetric Poisson matrix $L$ and scalar valued $S$ have to satisfy. A similar requirement is on $M$ and $E$. It is important to understand how easy it is to satisfy those, especially if the latent system is a genuinely high-dimensional one (meaning that different degrees of freedom are coupled). Let me give a specific example: the Hamiltonian dynamics in Remark 3.1 correspond to $S=0$ and $M=\textbf{0}$ (matrix), so the degeneracy condition is satisfied, but this is a trivial case for which one doesn't even need GENERIC. What if none of $S$, $E$, $M$, $L$ is zero?"
> > > > > > > > This question is about the mathematical formulation. Its answer should not be about machine precision, your algorithm, or the fact that there are 950+ references. Could you give three fundamentally distinct examples of physically relevant and genuinely high-dimensional systems, in which $S$, $E$, $M$ and $L$ are all nontrivial?
> > > > > > > >
> > > > > > > > - In my 2nd round I wrote "my suggestion on reducing the error and bias of the list of existing work was perhaps missed, but I still think this is important." I already gave concrete suggestions in the 1st round, and these suggestions can certainly be discussed, but I haven't seen any response in all the rounds.

---

> > > > > > > > > ### Author Response · Authors · 2021-08-25
> > > > > > > > > **Response**
> > > > > > > > >
> > > > > > > > > We'd first like to thank the reviewer for their patience. As pointed out, we had misinterpreted the question as being about whether the proposed scheme works in high dimensions, and missed that the reviewers focus was regarding whether there are examples of GENERIC/metriplectic systems in high dimensions. We provide three examples of problems where we think this framework will make a large impact.
> > > > > > > > > - The Navier-Stokes equations governing fluid flow admit this form. In complex fluids + soft matter, Pep Espanol developed a theory for coarse grained molecular dynamics called Smoothed Dissipative Particle Dynamics (SDPD) which describes thermally fluctuating materials (Espanol, Pep, and Mariano Revenga. "Smoothed dissipative particle dynamics." Physical Review E 67.2 (2003): 026705.). In this scheme, one works with an additive energies/entropies of the form E = \sum_i E_i, which assumes each particle is governed by a single energy/entropy E_i,S_i, which depend on the local particle state (thermodynamic variables like density, pressure, internal energy). People regularly simulate billions of particles with this scheme (there is an HPC implementation in the LAMMPS MD simulator). However, application of this theory is restricted to relatively simple models for constitutive closure (e.g. for non-Newtonian fluids, how does one model the coupling between micro and macrostructure). The data-driven approach advocated here provides a means to learn more complicated closures from both high-fidelity molecular simulation and rheometry experiments, and we are working on a follow up paper along these lines now.
> > > > > > > > > - In climate, conservation laws governing flow in the atmosphere also take this form (https://arxiv.org/pdf/2102.08293.pdf). There is an ongoing effort in the Energy Exascale Earth Systems project (E3SM.org) to use these improved models as the foundation for the atmosphere submodel of the global climate model. This is about as big as simulations can get at the moment (exascale refers to codes running 10^18 floating point operations per second), with ~trillion degrees of freedom. In this setting, extreme care has to be taken with the discretization of the governing PDE to obtain an ODE in generic form, but this has benefits in terms of forecasting long term predictions in climate, particularly as chaotic systems are classically ill-posed due to the "butterfly effect" and structure preservation is critical to ensuring physically meaningful long term predictions. In climate, these models require complicated closures to model physics below the available resolution (even at these massive scales one cannot resolve turbulence, for example) and processes involving multiphase flow/chemistry/phase transition such as modeling of clouds have been identified as a "priority research direction" for incorporating AI/ML closures into these massive scale codes (see e.g. https://science.osti.gov/wdts/scgsr/How-to-Apply/Priority-SC-Research-Areas#BER%20f).
> > > > > > > > > - In plasma physics governing the development of fusion energy along with astrophysical phenomena, it is particularly challenging how to model so-called magnetohydrodynamics which describe how the turbulent fluid flow of ions is coupled to electromagnetic (E&M) phenomena through Maxwell's equations. For this problem, one is faced with the question of coupling all of the fluid modes governing turbulence (i.e. resolving the Kolmogorov spectrum of turbulence, which is quite large). Unlike classical turbulence, Maxwell's equations lead to a nearly intractable closure problem: eddies generate an electromagnetic field which couples all scales, unlike classical fluids in which only modes of similar length scales are coupled. This too has metriplectic/GENERIC structure (https://michael-kraus.org/publications/mi_landau.pdf). Data-driven modeling is extremely attractive for this problem, where turbulence modeling is less mature compared to traditional single-phase turbulence.
> > > > > > > > >
> > > > > > > > > Your suggestions for references are excellent and we have added those to the manuscript.

---

> > > > > > > > > > ### Comment · Reviewer_AB4a · 2021-08-25
> > > > > > > > > > **Thanks for the examples**
> > > > > > > > > >
> > > > > > > > > > My concerns are finally addressed. The examples you gave are physically very important ones, and they help the readers appreciate the work. I increased my rating to 7.

---

### Official Review · Reviewer_D146 · 2021-07-16

**Rating:** 5
**Confidence:** 3

**Summary:**

This paper tries to learn an irreversible dynamic system with dissipation that has entropy growth. The author used a rigorous treatment and parameterization for the dissipation part and separated it from the reversible part. The parameterization naturally satisfies the degeneracy conditions. The author shows that compared to other methods that try to learn degeneracy, like the penalty-based method, this approach preserves the dynamics of damped systems in the long run.

**Limitations And Societal Impact:**

I doubt the generalization of this work as the experiments are relatively simple damped systems. I suspect the treatment in this literature can forecast some more chaotic systems or predicting something like the “butterfly effect.” There have been some potential impacts, but currently, I can only see it works for some well-studied fundamental problems.

**Main Review:**

Originality: Preserve metriplectic structure is pretty novel compared to the Hamiltonian system, and this parametrization using the GENERIC formalism is new.

Quality and Clarity: The theory and math are correct and well written. The experiments are not so appealing and convincing, although some comparisons are provided. The selection of the non-observable state is arbitrary, and no physical interpretation is provided in the paper.

Significance: The significance is not vital as some pioneering work by neural ODE and penalty-based method for dissipation treatment. This work has a more complex network while showing some noticeable improvement. In the real world, learning entropy growth is not as exciting and valuable as learning energy conservation.

**Time Spent Reviewing:**

2 hours

---

> ### Author Response · Authors · 2021-08-10
> **Response to Reviewer D146**
>
> We thank the reviewer for taking the time to read and provide thoughtful feedback of our paper. We respond point by point to the reviewers suggestions below. Our response to quotes are prefaced by (--)
>
> - " Quality and Clarity: The theory and math are correct and well written. The experiments are not so appealing and convincing, although some comparisons are provided. The selection of the non-observable state is arbitrary, and no physical interpretation is provided in the paper."
> --- As elaborated further in the response to reviewer AB4a, our intent in this work was to establish our theoretical framework which guarantees enforcement of the 1st+2nd laws of thermodynamics in addition to a fluctuation-dissipation theorem, and then use simple systems with analytic solutions for the entropic variable to allow a simple yet complete comparison. As discussed in AB4a, these types of comparisons served as a basis of the simpler reversible paper accepted to neurIPS ("Hamiltonian Neural Networks" Greydanus et al, NeurIPS 2019), and we chose to pursue similarly simple example for pedagogical reasons and avoid application specific complications. The numerical results do in fact confirm the theoretical results, and we have listed in the response to reviewer AB4a a survey of the diverse set of scientific applications for which this type of structure will make a broad impact. In particular, these toy problems serve to indicate that even in simple settings, the current state-of-the-art (physics-informed networks with enforcement of physics by penalty/regularizer) is insufficient to provide the type of physical-realizability guarantees which are standard in science and engineering.
> --- We are unclear on the critique that "the selection of the non-observable state is arbitrary". In classical statistical physics, there is no unique choice of an entropic variable. For example, one may choose to work with a specific entropy, an internal energy or enthalpy, or a temperature - all of which provide a characterization of the energy contained in microscopic scales. Informally, the variable s_i may be understood as providing a thermal reservoir which stores dissipated energy. The specific choice or interpretation of s_i is unimportant - so long as it is extracting the correct amount of energy from the system, the process is consistent with the second law of thermodynamics, and (in the case of thermally forced systems) the work done by thermal fluctuations is exactly balanced by that dissipated.
>
> - "Significance: The significance is not vital as some pioneering work by neural ODE and penalty-based method for dissipation treatment. This work has a more complex network while showing some noticeable improvement. In the real world, learning entropy growth is not as exciting and valuable as learning energy conservation."
> --- We respectfully disagree with the reviewer on this point, although we acknowledge that everyone may have personal preferences regarding what they find exciting :) . We first point out that our framework reduces to learning energy conservation in the case where S = 0. Therefore "learning energy conservation" is a proper subset of the class of models which may be handled in this framework. Secondly, from a physical perspective, the vast majority of physically relevant systems are not reversible, beyond those taught in an introductory undergraduate physics coarse. For example, Hamiltonian/Lagrangian networks focusing on energy conservation have shown promise for learning data-driven models for control of robotics (Cranmer, Miles, Sam Greydanus, Stephan Hoyer, Peter Battaglia, David Spergel, and Shirley Ho. "Lagrangian neural networks." arXiv preprint arXiv:2003.04630 (2020).). In reality, the joints involved in robotics undergo friction, particularly over time as parts undergo wear, and the development of AI-enabled "digital twins" which can adaptively correct for wear and tear has been identified as a priority research direction by funding agencies (Baker, N., Alexander, F., Bremer, T., Hagberg, A., Kevrekidis, Y., Najm, H., ... & Lee, S. (2019). Workshop report on basic research needs for scientific machine learning: Core technologies for artificial intelligence. USDOE Office of Science (SC), Washington, DC (United States).). Another major interest is using data-driven dynamics to obtain reduced order descriptions of systems (Afkham, Babak Maboudi, and Jan S. Hesthaven. "Structure preserving model reduction of parametric Hamiltonian systems." SIAM Journal on Scientific Computing 39, no. 6 (2017): A2616-A2644.). For these problems, it has been known since the 60s that coarse-graining of even reversible dynamics requires introducing of dissipation to account for memory effects (Chorin, Alexandre J., Ole H. Hald, and Raz Kupferman. "Optimal prediction and the Mori–Zwanzig representation of irreversible processes." Proceedings of the National Academy of Sciences 97, no. 7 (2000): 2968-2973., Mori, H., 1965. Transport, collective motion, and Brownian motion. Progress of theoretical physics, 33(3), pp.423-455.). Further, for dynamics occurring at the mesoscale (i.e. nanometer to micron range) it was established in Einstein's work as early as 1905 (Einstein, A., 1956. Investigations on the Theory of the Brownian Movement. Courier Corporation.) and common knowledge in 1966 (Kubo, Rep. "The fluctuation-dissipation theorem." Reports on progress in physics 29.1 (1966): 255.) that microscopic transport processes depend critically upon the interplay of dissipative/entropic effects and thermal noise. These concepts from modern physics govern diverse applications: aerodynamics (Parish, Eric J., and Karthik Duraisamy. "A dynamic subgrid scale model for large eddy simulations based on the Mori–Zwanzig formalism." Journal of Computational Physics 349 (2017): 154-175.), chemical reaction/combustion kinetics (Donev, A., Nonaka, A., Sun, Y., Fai, T., Garcia, A. and Bell, J., 2014. Low Mach number fluctuating hydrodynamics of diffusively mixing fluids. Communications in Applied Mathematics and Computational Science, 9(1), pp.47-105.), biomechanics (Wang, Q., Ripamonti, N. and Hesthaven, J.S., 2020. Recurrent neural network closure of parametric POD-Galerkin reduced-order models based on the Mori-Zwanzig formalism. Journal of Computational Physics, 410, p.109402.), social dynamics (Short, Martin B., P. Jeffrey Brantingham, Andrea L. Bertozzi, and George E. Tita. "Dissipation and displacement of hotspots in reaction-diffusion models of crime." Proceedings of the National Academy of Sciences 107, no. 9 (2010): 3961-3965.), and the power grid (RRoth, Jacob, David A. Barajas-Solano, Panos Stinis, Jonathan Weare, and Mihai Anitescu. "A kinetic monte carlo approach for simulating cascading transmission line failure." arXiv preprint arXiv:1912.08081 (2019).), to name a few.
>
> - "There have been some potential impacts, but currently, I can only see it works for some well-studied fundamental problems."
> --- As noted in the previous example, there are a large number of examples where the proposed structure preservation is shown to be critical for providing predictive and robust (i.e. numerically stable) simulations - please see for example the nice review (Öttinger, H. C. (2018). GENERIC: Review of successful applications and a challenge for the future. arXiv preprint arXiv:1810.08470.) along with the 950+ references for the textbook (Öttinger, Hans Christian. Beyond equilibrium thermodynamics. John Wiley & Sons, 2005.).
>
> -"The significance is not vital as some pioneering work by neural ODE and penalty-based method for dissipation treatment."
> --- Without specific feedback on what the reviewer views as vital about neural ODE, and references to specific papers it is difficult for us to specifically comment, but we hope that the list of references we've provided above conveys a bit about why we are so excited about this line of research and optimistic for it to majorly impact scientific machine learning problems across a range of disciplines.

---

### Decision · Program_Chairs · 2021-09-28

**Decision:**

Accept (Poster)

**Comment:**

This paper underwent an involved discussion that lead one reviewer to first decrease (6 to 5) and then increase (5 to 7) their score.
After the discussion, one reviewer is still strongly convinced about rejection, providing as main criticism insufficient empirical evidence.

**Consistency Experiment:**

NeurIPS has a long history of experimentation. In 2014, NeurIPS ran an experiment in which 10% of submissions were reviewed by two independent committees to quantify the randomness in the review process. This year, we repeated a variant of this experiment to see how the quality of the review process has changed over time.  This paper was part of the experiment and was therefore assigned to two committees (consisting of reviewers, an Area Chair, and a Senior Area Chair) that reached independent decisions.  If both committees made the same recommendation, this recommendation was followed. If a single committee recommended acceptance, the paper was accepted (with the exception of a few cases in which the other committee identified what we considered a fatal flaw, e.g., an error in a key result).

This copy’s committee reached the following decision: **Reject**

The other committee assigned to the paper recommended **Accept (Spotlight)**.  You can find the other set of reviews, along with any follow up discussion with the authors here:
https://openreview.net/forum?id=ntAkYRaIfox